# A pilot study combining noninvasive spinal neuromodulation and activity-based neurorehabilitation therapy in children with cerebral palsy

Susan Hastings[1], Hui Zhong[2], Rochel Feinstein[3], Gittel Zelczer[3], Christel Mitrovich [4], Parag Gad [2,5] ✉ & V. Reggie Edgerton [2,5,6,7]

Cerebral Palsy (CP) is the most common pediatric motor disability with multiple symptoms and etiologies. CP is exhibited through sensorimotor delays, impaired posture resulting in limited activities and participation. Our recently concluded, single arm, unblinded, pilot study (NCT04882592) explored whether an intervention combining non-invasive spinal neuromodulation during an activity-based neurorehabilitation therapy (ABNT) can improve voluntary sensory-motor function captured via the Gross Motor Function Measure (GMFM-88) scores (primary outcome). Sixteen children diagnosed with CP with Gross Motor Function Classification Scale levels I-V were recruited and received the same intervention (2x/week for 8 weeks) to correct the dysfunctional connectivity between supraspinal and spinal networks using the normally developed proprioception. We demonstrate that the intervention was associated with clinically and statistically significant improvement in GMFM-88 scores in all children, thus meeting the prespecified primary endpoint. However, the improvement with ABNT alone needs further exploration. No serious adverse events were observed (safety endpoint).

Cerebral palsy (CP) is an umbrella diagnosis that is the most common motor disability in childhood and impacts 1.5 to 4 of every 1000 births[1–3], leading to over 10,000 new cases diagnosed each year. More than 500,000 children under the age of 18 currently have CP in USA, with over 17 million cases worldwide. The more common pathology that leads to CP is among supraspinal networks and could be due to multiple etiologies. In some cases, these supraspinal pathologies also manifest as motor neuronal dysfunctions, perhaps due to aberrant supraspinal-spinal connections formed early in life[4]. If functionally abnormal connections persist significantly beyond the early developmental phase, the supraspinal dysfunction can impose dysfunctional

connections among the normally developed spinal networks, thus generating abnormal sensory-motor responses[5]. Without interventions, these functionally abnormal synaptic connections are reinforced further during early childhood, further contributing to the neuromuscular disorders associated with CP, such as neuromuscular spasticity, balance deficits and poor coordination among motor pools. Children with CP with relatively few functional limitations have been reported to be substantially weaker than typically developing peers[6]. Presently, the predominant goals of current CP treatment options are to manage symptoms, relieve pain, and maximize independence to achieve a long life, despite having irreversibly debilitating conditions.

[1]Susan Hastings Pediatric Physical Therapy, San Jose, CA 95128, USA. [2]Rancho Research Institute, Downey, CA 90242, USA. [3]OPTimal care Therapy, Lakewood, NJ 08701, USA. [4]Reneu Health, San Diego, CA 92111, USA. [5]SpineX Inc., Los Angeles, CA 90064, USA. [6]USC Neurorestoration Center, University of Southern California, Los Angeles, CA 90033, USA. [7]Institut Guttmann. Hospital de Neurorehabilitació, Institut Universitari adscrit a la Universitat Autònoma de Barcelona, Barcelona, 08916 Badalona, Spain. ✉e-mail: parag@spinex.co

**Fig. 1 | Overall Experimental Design. a** Experimental setup demonstrating the placement of non-invasive electrodes over the cervical and thoracic regions of the spinal cord while the child is actively engaged in activity based neurorehabilitation therapies, **b** Experimental design for each child including initial screening and recruitment period, 16 training sessions (red arrows) over the 8 weeks and baseline and final assessments.

Other methods to treat spasticity associated with CP include intramuscular Botulinum ToxinA injections, medications such as Baclofen, surgical interventions such Selective Dorsal Rhizotomies (SDR) and orthopedic surgeries[7]. While these methods may provide a short-term reduction in spasticity, in the long term, they may negatively impact functional changes, especially in a growing child.

Standard of Care (SoC) treatment for CP often include one or more of the following: 1) an activity-based neurorehabilitation therapy (ABNT) play-based approach, appropriate for age, with therapist facilitating more normal movement patterns with the goal of strengthening selected groups of muscles; 2) direct orthopedic strengthening exercises, often used in combination with stretching to maintain range of motion, and 3) increasing activity participation by emphasizing on movement for sustaining metabolic and cardiovascular fitness and maintaining the ability of the child to move and function to his/her best ability, and become more independent through utilizing devices to help them sit, stand, or walk. Historically, the change in Gross Motor Function Measure (GMFM-88), a gold standard clinical instrument to measure voluntary sensorimotor function scores, observed for currently used interventions ranges from 1.3 to 13.4 points over the course of 3 weeks to 18 months of treatment[8,9] with potential decreases after 18 to 36 months[10]. However, most of these interventions are designed to decrease spasticity, while having little effect in improving GMFM-88 scores and in acquiring an expanded range of self-initiated and well-controlled movements.

Based on our pre-clinical and clinical studies after spinal cord injury (SCI), we have identified spinal neuromodulation strategies in conjunction with engaging the proprioception generated during ABNT to be critical components of the overall control of posture and locomotion[11,12]. Over the last few years we have developed and evaluated a pediatric version of the noninvasive spinal neuromodulation

modality called SCiP™ (SpineX Inc., Los Angeles, CA) (Fig. 1). Previously, we have demonstrated that noninvasive spinal neuromodulation can lead to recovery of lower extremity[13–15], upper extremity[16,17], and trunk function[18], as well as bladder and bowel[19–23] and breathing[24] function after SCI. However, the use of noninvasive spinal neuromodulation during ABNT in CP has not been studied. Here we have used a combination of an activity-based neurorehabilitation therapy (ABNT) modality consisting child selected, play-based activities while successfully leveraging optimal external assistance, weight shifts and alignment of the body's center of mass to accommodate gravitational vectors as needed to sustain equilibrium. These training methods are in line with the recent guidelines and recommendations from the Cerebral Palsy Alliance (CPA) and addresses all the necessary components defined by the World Health Organization's (WHO) International Classification of Function framework for CP wherein, treatment is focused on activity, participation, environmental factors that can help the child's function, in addition to personal factors, such as allowing the child to participate in an activity of interest. In the current study, the children were treated with ABNT appropriate to their age level, while participating in a self-chosen activity of interest to the child, allowing the child to play while interacting with family and friends[25]. The primary objective of our study was to determine whether the combined effect of noninvasive spinal neuromodulation[15,18,26] and our ABNT paradigm is associated with increased GMFM-88 scores.

## Results

All children demonstrated GMFM-88 scores consistent for their age and GMFCS level at the start of the study period with scores ranging from 68 to 85 for levels I and II and 2.9 to 29 for levels IV and V (historically reported scores: level. I, $76.29 \pm 17.14$, level II, $59.51 \pm 13.49$, level IV $30.21 \pm 10.79$, and level $8.33 \pm 5.16$[27]). Positive responses were observed in all children during the first ABNT session with noninvasive spinal neuromodulation including improvement in posture and sensorimotor capabilities consistent with previous results[26]. For individuals with levels, I and II, improvements included the ability to take more independent steps with their center of mass (CoM) sufficient to sustain equilibrium, upright posture and improved balance. Children with level IV and V demonstrated more subtle improvements including better voluntary head control, ability to sit for longer durations without external support and lower assistance needed during stepping. Notably, all children became more interested in the chosen activity and often responded positively via facial expressions when successfully interacting with cause-and-effect toys. This was observed in almost all the children at Levels IV and V when previously, they did not seem interested in interacting with any toys.

Over the course of 8 weeks of training every one of these children demonstrated clinically significant improvement in their GMFM-88 scores (minimal clinical important difference, MCID = 5 points[28]). Overall, the change in GMFM-88 score (primary outcome) was also statistically significant ($P < 0.05$) (Fig. 2a). Children initially diagnosed as GMFCS levels I and II improved from $70.4 \pm 4.13$ to $85.28 \pm 2.7$ ($P < 0.05$), these improvements are considered to be clinically significant with all I and II children considered to be responders. In addition, children initially diagnosed as GMFCS levels IV and V showed clinical improvement in their GMFM-88 scores from $11.09 \pm 3.10$ to $21.92 \pm 3.85$ ($P < 0.05$) (Fig. 2). The improvements in GMFM-88 scores were accompanied by spontaneous emergence of new sensorimotor skills, as well as improvements in previously learned skills (supplementary video 1). At the start of therapy, 9 children were nonambulatory (needed maximal external assistance to take steps overground, GMFCS level IV, $n = 3$ and level V, $n = 6$) and needed assistance while stepping whereas 7 were able to take some steps (GMFCS level I and II, $n = 7$). At the end of 8 weeks of therapy, only 4 children were nonambulatory (GMFCS level IV, n = 0 and level V, n = 4), 3 were capable to stepping with minimal assistance (GMFCS level IV, n = 2 and

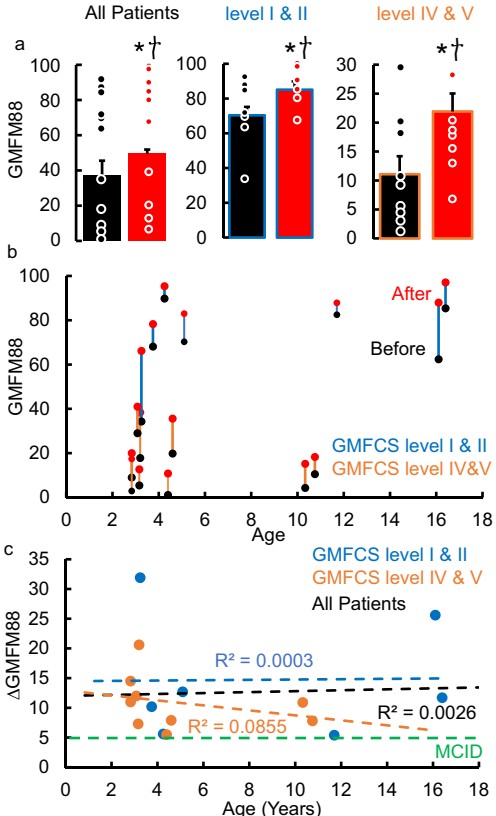

**Fig. 2 | Primary outcome changes. a** mean ± SE (*n* = 16) GMFM-88 scores before and after therapy; mean ± SE (*n* = 7) GMFM-88 scores before (black) and after (red) therapy for GMFCS levels I and II (blue) and; mean ± SE (n = 9) GMFM-88 scores before (black) and after (red) therapy for GMFCS levels IV and V (orange). **b** GMFM88 scores at the start (black) and end of therapy (red) relative to age of the child at the start of the therapy and **c** ΔGMFM-88 scores relative to their age at the start of the therapy. All data were tested for normality using the Kolmogorov-Smirnov test. Based on the result of a normal distribution, paired t-tests were used to compare the group mean data before and after therapy. * Significantly different from before therapy at *P* < 0.05. † Meaningful clinically improvement difference (MCID) = 5 points. Source data are provided as a Source Data file.

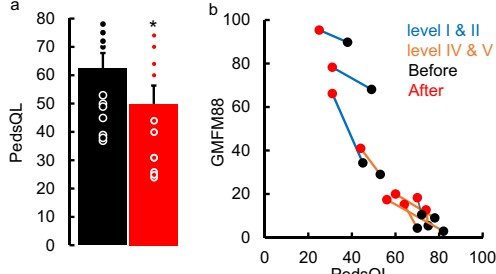

**Fig. 3 | Quality of Life Changes. a** mean ± SE (n = 12, GMFCS levels I and II, n = 3, GMFCS levels IV and V, n = 9) PedsQL scores before and after therapy. All data were tested for normality using the Kolmogorov-Smirnov test. Based on the result of a normal distribution, paired t-tests were used to compare the group mean data before and after therapy. * Significantly different from before therapy at *P* < 0.05. Note the max score on the PedsQL is 88 points and a decrease in score represents an improvement in function. **b** X-Y plot for correlating GMFM-88 scores at start and end of therapy with the PedsQL scores demonstrating that the increase in GMFM-88 scores (improved sensorimotor function of the child) is directly linked to the decrease in PedsQL scores (improved quality of life of parent). Please note, data for *n* = 12 out of 16 are shown since 4 PedsQL data points were not collected. Source data are provided as a Source Data file.

level V, n = 1) and 10 were able to step independently with no external assistance (GMFCS level I and II, n = 7, level IV, n = 1 and level V, n = 1) (primary outcome). In addition, during interviews with the study team, all parents reported that after two to four treatment sessions, their child voluntarily began to practice the newly learned motor skills at home, even in the absence of neuromodulation, thus further enhancing neuroplasticity and increasing the child's activity level. Parents and caregivers also reported that their children demonstrated improvement in many of the symptoms secondary to cerebral palsy resulting in significant improvement in their quality of life as measured by the PedsQoL survey (Fig. 3).

## Discussion

We demonstrated the effect of noninvasive spinal neuromodulation during ABNT over a short period of time (8 weeks) with the improvement in GMFM-88 scores being higher in magnitude (>3x) than the minimal clinically important difference (MCID = 5 points[28]) and every child being considered as responders.

There are several limitations in this first in human, single-arm, unblinded pilot study. While the key objective of the current study was to determine the combined effect of ABNT and noninvasive spinal neuromodulation on the primary outcome (GMFM-88), the independent effects of ABNT and spinal neuromodulation alone and the effect of sham neuromodulation needs to be evaluated in future blinded studies. In addition, larger randomized clinical trials with appropriate control groups including standard of care therapies such as physical therapy, pharmacological therapy to manage spasticity etc., are needed to better predict the magnitude of change in each age group and each level of severity. The current study does not include GMFCS level III and the effectiveness of the interventions for all age groups remains uncertain. In addition, the long-term (over the course of several years) impact of spinal neuromodulation and ABNT on activity participation, joint health, potential deformities and induced pain remains unknown. Based on our hypothesis and experience to date with studying the combined effects of noninvasive spinal neuromodulation during ABNT in individuals living with SCI[11], we hypothesized that it may be possible to sustain the level of improvement in function for up to 1 to 2 months after the termination of the intervention, although this remains to be tested in this context. It seems likely, however, that one of the factors that may define this rate of loss of function after the intervention is stopped is dependent on the child sustaining a critical level of activity post the intervention. While the present observations were derived from a heterogenous population based on levels of severity as well as age of the child, the results are encouraging given some improvements from baseline in sensorimotor functions were observed in all children, and these changes occurred within a relatively short period of time compared to other interventions. It is also important to note that children as young as two years up to 16 years of age and GMFCS levels I to V responded positively. Certain new skills and motor patterns, however, may take longer to correct in older and more involved individuals that have acquired certain inefficient motor patterns over a period of years.

Multiple attempts have been made to activate the nervous system such as using central and peripheral stimulation and combining it with robotic gait training leading to improvement in GMFM-88 scores in some of the participants[29]. Unlike traditional approaches (such as functional electrical stimulation, FES), which has been used most often to directly stimulate muscles, we have previously delivered modest stimulation below motor threshold[19,23,26,30] to neuromodulate the physiological states of the spinal networks, and perhaps even supraspinal networks. It appears that these neuromodulated states of spinal networks resulted in improved coordinated movements during stepping. Thus, as hypothesized, it appears that the relatively normal proprioceptive input provided a critical source of movement control of the intended task as shown in adults with SCI[31] and hypothesized for children with CP[4].

It appears that practicing motor tasks with incorrect alignment and posture could lead to enhancement of aberrant connections within and among the brain and spinal neural networks[32]. Based on our previously demonstrated acute neurophysiological results[26], we hypothesize that the physiological state of predominantly supraspinal aberrant connections can be suppressed with spinal neuromodulation to a level that allows the children to voluntarily initiate the intended movement, more normally, and thus progressively generate more normal patterns of proprioception derived from improved signals generated by the spinal networks[4]. We hypothesize that multisite spinal neuromodulation at the cervical and thoracic levels enables bidirectional communication with supraspinal centers[13,33], providing sufficient functional reorganization of supraspinal-spinal connections that can facilitate sensorimotor function. In essence, spinal neuro-modulation appears to facilitate neuroplasticity and leads to the for-mation of more normal bidirectional interactions of the Brain-Spinal Cord-Muscle-Spinal Cord-Brain networks.

Based on numerous studies combining electrical spinal neuro-modulation with task-specific training[4,14,15,17], we hypothesize that appropriately trained therapists can provide the corrective support by facilitating kinematic realignment that generates proprioceptive ensembles that accommodates gravitational vectors necessary for sustaining equilibrium. As hypothesized previously[4], the present data are consistent with the concept that the projection of these proprio-ceptive ensembles to spinal networks enables a normalizing of motor learning and network functional neural reorganization that can lead toward more effective movements. This synergistic transformation of functional connectivity between the spinal networks and the improved proprioception that the spinal networks receive. When the ABNT during noninvasive spinal neuromodulation is self-selected by the child, we observed more creative exploration in performing new motor skills (in pediatrics usually play related) rather than requiring a more standard, but less interesting activity for the child. The basic rationale of this approach is to generate a wider range of patterns of proprioception from a greater range of movements, from which spinal networks can experience and translate to different motor tasks. The-oretically, the interaction between the spinal motor output and pro-prioception can gradually minimize the pathological inputs from the injured brain in controlling posture and locomotion.

In conclusion the present results suggest that a combination of spinal neuromodulation and ABNT is associated with improvement in self-initiated sensorimotor functions as recorded on the clinically significant improvement in GMFM-88 scores. Further, these improved functions are associated with improvement in quality of life of the parents. Indirectly, the present results demonstrate noninvasive elec-trical spinal neuromodulation combined with an ABNT strategy may facilitate improved functional, bidirectional connectivity between spinal and supraspinal networks in individuals with CP. Finally, the present data are consistent with the concept that spinal networks that are normally developed in individuals with CP can be engaged by proprioception to serve as a source of control of movements

## Methods

The study was approved by an external Investigational Review Board (Advarra IRB). The study was listed on clinicaltrials.gov (NCT04882592). Sixteen children were recruited for this single arm, unblinded, non-randomized, prospective study between May 2021 (first enrollment) and February 2022 (last exit) (Table 1). All study participants' parents signed the informed consent form and consented to their data to be used for future publications and presentations. The participants did not receive compensation to be part of the study. The authors affirm that a parent of the participant provided consent for publication of the video in Supplementary video 1. The inclusion cri-teria included 1) individuals above the age of 2 years of age and 2) diagnosed with cerebral palsy (CP). The exclusion criteria included 1)

### Table 1 | Demographics of participants

| Patient ID | Age (years) | Gender | GMFCS Level | CP Diagnosis |
|---|---|---|---|---|
| 1 | 12-18 | M | I | Mixed |
| 2 | 7-12 | M | I | Spastic Biplegia |
| 3 | 2-7 | M | II | Spastic Biplegia |
| 4 | 2-7 | M | II | R Spastic Hemiplegia |
| 5 | 2-7 | M | II | Post Hemispherectomy |
| 6 | 12-18 | M | II | Spastic Quad |
| 7 | 2-7 | M | II | Spastic Biplegia |
| 8 | 2-7 | M | IV | Athetoid |
| 9 | 2-7 | M | IV | Spastic Quad |
| 10 | 2-7 | F | IV | Ataxic |
| 11 | 2-7 | F | V | Infantile |
| 12 | 2-7 | M | V | Spastic Quad |
| 13 | 7-12 | F | V | Dyskinetic Quad |
| 14 | 2-7 | M | V | Spastic Quad |
| 15 | 2-7 | M | V | Spastic Quad |
| 16 | 7-12 | F | V | Dyskinetic Quad |

Children's initial demographics and GMFCS levels. *GMFCS* Gross Motor Functional Classification Scale.

selective dorsal root rhizotomy, 2) intramuscular Botox injections or orthopedic surgeries in the preceding 12 months, 3) current antispastic medications, 4) unhealed fractures or contractures that would prevent them from performing functional tasks and 5) other experimental therapies that were judged to be conceptually inconsistent with the underlying neurophysiological hypothesis of the present experiment.

### Transcutaneous spinal neuromodulation

Spinal neuromodulation was delivered in the clinic using a proprietary SCiP™ device (SpineX, Inc). The stimulation waveform consists of two alternating pulses of opposite polarities separated by a 1uS delay forming a delayed biphasic waveform. The pulses consisted of a high-frequency biphasic carrier pulse (10KHz) combined with a low fre-quency (30 Hz) burst pulse each with a pulse width of 1 ms. Simulta-neous spinal neuromodulation was applied using two adhesive round electrodes (1.25" dia) located between C5-6 and T11-12 serving as the cathodes, and two adhesive rectangular electrodes (3×5") over bilateral iliac crests as common anodes for all children (Fig. 1). The children and parents were blinded from the exact intensity of stimulation for an unbiased outcome. Based on our previous studies that used electro-myography (EMG) as a biomarker to determine motor threshold[26], during training, a sub-motor threshold intensity (20% below lowest lower extremity muscle motor threshold) was used to ensure no motor evoked responses are generated (any lower extremity muscle contrac-tions) and to ensure no pain or discomfort is being caused to the child. In this study, the threshold was defined as the amplitude at which the child first attempted to extend the cervical or thoracic regions while in a seated position. Note, since the stimulation electrodes were placed at C5-6 and T11-12, the stimulation thresholds were significantly lower compared to our previous study with electrodes over T11-12 and L1-2. Once thresholds were determined (range: C5-6: 18-22 mA, and T11-12: 16-20 mA) the stimulation intensity was set 20% below this threshold. The intensities over the C5-6 spine ranged between 12 and 18 mA and over the T11-12 ranged between 10 to 16 mA based on the activity being performed. During activities involving sitting, rolling etc. amplitudes were further lowered by 1–2 mA, whereas, during standing and step-ping, the intensities were increased by 2–4 mA. During the course of a given activity, the intensities would be modulated ±2 mA based on observed functional performance of the child. In addition, the children often provided feedback regarding the intensity of stimulation and if it

was sufficient or not based on the activity being performed. These procedures are routinely performed and consistently found to be excellent subject-specific input in our lab with patients with spinal cord injury, stroke, multiple sclerosis and cerebral palsy.

All children were able to communicate with the parents and research team if they experienced any pain or discomfort either due to the spinal neuromodulation or due to the locomotor procedures. None of the children reported any pain or discomfort during the neuromodulation and were blinded from the neuromodulation parameters at any given time during the testing procedures. While the children could feel the stimulation pulses initially, they were unable to clearly distinguish between amplitudes or even the presence of stimulation after a brief period of accommodation. All children tolerated the neuromodulation well and did not report any pain or discomfort. In children that are non-verbal, facial expressions were observed closely during treatment sessions, and stimulation intensity was consistently reduced if they became irritable or agitated. No adverse events were reported during the course of the study.

### Activity-based neurorehabilitation therapy (ABNT)

The children were asked to take off their shoes and braces (if any) before each therapy session. Each therapy session began with 1) treadmill training to provide maximum proprioceptive input to prime the nervous system. The treadmill speeds ranged from 0.1 m/s to 0.5 m/s based on the child's capability. Assistance was provided as needed by therapist(s) including i) moving limbs while stepping with the goal being to accurate place the heel and toes on the treadmill belt and achieve consistency of gait, ii) appropriate weight-shifting at the pelvis and to advance their lower extremity during swing out in front of the Center of Mass and iii) maintain the head, shoulders, hips and heels in alignment. Next, ABNT included functional activities during play activities and walking with body support for alignment (as needed) including, 2) upright sitting, with trunk control and weight shifting during reaching, 3) transitions including sit to stand, rolling prone to a quadruped position, consecutive rolling etc., 4) dynamic standing with postural and weight shifts, 5) side-stepping with frontal plane weight shifts, 6) backward stepping, 7) Climbing/creeping up an incline, 8) climbing up and down steps and 9) over ground walking. Children classified as GMFCS levels I and II underwent all the components of the therapy listed above, whereas those classified as GMFCS levels IV and V focused more on activities 1-6. All children were trained 2 sessions a week for 8 weeks with each session lasting for 50-60 mins. In case a child missed a session, a make-up session was performed with the next 7 days to ensure completion of 16 sessions within 8 weeks. During all therapeutic activities, it was ensured that appropriate posture, midline orientation and head position was maintained with maximal weight-bearing to optimize proprioceptive information being processed by the nervous system. Since children with GMFCS levels I and II are functional, but with limited motor patterns available, the objective was to enable self-generated movement patterns with more typical posture and alignment, whereas lesser functioning children with GMFCS levels IV and V, the objective was to allow for new motor patterns to evolve during play activities, while retraining existing patterns by using proper posture and alignment. Toys with lights and sounds were used to engage children to facilitate movements including reaching, grasping and visual tracking etc. All children were asked to discontinue other therapies, except for aqua and hippotherapy, prior to initiation of and during the study[34]. Both aqua and hippo therapies are typical activities for children, and they do not place excessive motor demands on the child.

### Clinical and functional assessments

All assessments were completed within a single session before and after the 8 weeks of training sessions in the absence of spinal neuromodulation and were done by an experienced and board-certified pediatric physical therapist. The team members remained unblinded during the study. The primary outcomes included 1) GMFM-88 scores and 2) qualitative assessment of stepping on the treadmill with minimal support and assistance, as needed) and qualitative assessment of stepping overground (with minimal support and assistance, as needed). The GMFM-88 is an 88-item measure assessing gross motor activities across five dimensions: A) lying and rolling, B) sitting, C) crawling and kneeling, D) standing, and E) walking, running and jumping and is considered the gold standard in the U.S. for measuring gross motor function change over time in children with CP. Children were designated as responders if they realized at least a 5-point increase in GMFM-88 score from baseline. Stepping assessments addressed three specific criteria; the ability to take weight-bearing steps independently, the ability to take steps with some assistance (assistance for weight bearing and/or assistance to move limbs) and the ability to take steps with full assistance (assistance for weight bearing and assistance to move limbs). In addition, parents/caregivers completed a quality-of-life satisfaction survey (PedsQL survey for CP,[35]) before and after therapy. The primary safety endpoint was the proportion of children who experience one or more serious adverse events (SAEs).

### Statistical analysis

All data are reported as mean ± SE. All data were tested for normality using the Kolmogorov-Smirnov test. Based on the result of a normal distribution, paired t-tests were used to compare the group mean data before and after therapy. All statistical significance is reported at $P < 0.05$. Since this was our first in human pilot study, power calculations were not completed prior to study initiation. The study is in compliance with ICMJE guidelines on reporting.

### Reporting summary

Further information on research design is available in the Nature Research Reporting Summary linked to this article.

## Data availability

Additional video data from training and testing sessions are available. Data can be made available on reasonable request by qualified investigators. Source data are provided with this paper.

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

## Acknowledgements

We would like to acknowledge all the children, their parents and families for their participation in the study. This research was funded in part by NIH/NIDDK R44DK129164, the Dana & Albert R. Broccoli Charitable Foundation, Nanette and Burt Forester, including matching by PwC LLP, Roberta Wilson, BEL13VE in Miracles Jack Jablonski Foundation, Consortium for Technology & Innovation in Pediatrics, Brain Recovery Project (BRP), Cerebral Palsy Alliance (Australia) and Cerebral Palsy Alliance Research Foundation (USA). The study sponsor had no role in study design, data collection, analysis or writing.

## Author contributions

S.H. Designed the study, designed the ABNT protocol, oversaw the study, performed all assessments, determined neuromodulation parameters, edited and approved the manuscript. H.Z.: Delivered the ABNT protocol and spinal neuromodulation, edited and approved the manuscript. R.F.: Delivered the ABNT protocol and spinal neuromodulation, edited and approved the manuscript G.Z.: Delivered the ABNT protocol and spinal neuromodulation, edited and approved the manuscript. CM: Delivered the ABNT protocol and spinal neuromodulation, edited and approved the manuscript. P.G.: Designed the study, analyzed the data, wrote, edited and approved the manuscript. V.R.E.: Designed the study, analyzed the data, wrote, edited and approved the manuscript.

## Competing interests

VRE has shareholder interest in Onward and SpineX. PG has shareholder interest in SpineX. All other authors declare no other competing interest.
