## [Peer Review File · Nature Communications]

A pilot study combining noninvasive spinal neuromodulation and a novel activity-based neurorehabilitation therapy in children with cerebral palsyREVIEWER COMMENTS

Reviewer #1 (Remarks to the Author):

This manuscript explores an intervention that combines activity-based neurorehabilitation therapy (ABNT) with transcutaneous spinal stimulation on gross motor function in children with CP. Children with CP were recruited to undertake an 8-week intervention of spinal stimulation combined with ABNT. Before and after the intervention, outcome measures (gross motor function scale and a pediatric quality of life questionnaire) were conducted.

Interest in the application of transcutaneous spinal stimulation following spinal cord injury has gained considerable momentum in the past few years. The application to children with CP is relatively novel and interesting. The authors have conducted an important clinical trial, worthy of publication, however there are several major issues in the manuscript that need to be addressed prior to publication. The main issue is that the authors make broad and categorical statements and claims throughout the manuscript, with insufficient evidence (either from their own results or the literature) to support them. In particular, there are several claims throughout the discussion and conclusion that are not supported by the results of the study. The discussion and conclusion need to be rewritten with a clear distinction between the findings of the study and speculation about the mechanisms behind the results, which the authors have not attempted to measure in the present study.

In the introduction, the authors provide insufficient background on standard of care, and the limitations of current interventions. In particular, the authors need to provide a more detailed description of ABNT, if and how this differs from standard of care, and evidence to support the effectiveness of ABNT given alone. Indeed, throughout the manuscript, the authors do not adequately address that they are exploring a combination of two interventions, with no control group, so it is not possible for them to separate the effects of each intervention, nor to say whether similar results would have been achieved with ABNT or spinal stimulation given alone.

The objectives stated at the end of the introduction are not met by the outcome measures, and the results of all outcome measures are not presented in the results.

One major point is that substantially more detail is required about the stimulation parameters used, how they were determined and their justification (see points below under methods).

Please modify the Title to better reflect the results presented in the manuscript.

Abstract

Lines 9-11: there is insufficient evidence to support the statement "Spinal neuromodulation and activity-based rehabilitation triggers neural network reorganization". Please modify.

Lines 17-18: the results presented in the paper do not support the statement "We demonstrate that transcutaneous spinal neuromodulation during ABNT can transform the neural networks in children diagnosed with CP to improve voluntary postural and locomotor activity". No neurophysiology or imaging was done to explore whether changes in neural networks occurred. In addition, no control group was included. Please replace this statement with the actual results that are presented in the paper.

Introduction:

Lines 47-48: "humans that have lost proprioception as an adult are functionally paralyzed, even though all descending motor pathways remain intact^{22,23}" The references provided do not support the statement. The studies by Gordon et al. and Ghez et al. (1995) present findings on people with proprioceptive deficits due to large-fibre sensory neuropathies. They report that that the patients made large spatial errors in movement, compared with controls, and that their accuracy improved with

visual feedback. The studies do not suggest that these patients were unable to move or “functionally paralyzed”. I would suggest the authors remove the statement unless they can provide appropriate references to support it.

Lines 48-49: “spinal networks can readily learn to perform a new motor task without input from the brain^{24,25}” Please clarify that this is true in Mammals but not in humans, as evidenced by several Locomotor Treadmill Training trials in humans with SCI.

Lines 49-50: “neuro plasticity can occur in spinal and supraspinal networks with spinal neuromodulation combined with ABNT^{26,27}” Again, the statement is not supported by the references. These studies show that neuromodulation of lumbosacral networks enables specific functional movements, but provide no direct evidence that neuroplasticity has taken place. In addition, both of these studies were conducted using implanted (epidural) electrodes, with no kHz carrier frequency, which is rather different to the neuromodulation being applied in the present study.

Line 54: Please define GMFM88, and provide references to support its validation in children with CP.

Methods:

Line 63: was there an upper age limit?

Table 1: Define GMFCS. Be consistent with decimal places.

Lines 74-77: Please provide justification for the parameters chosen. Recently, it has been reported that the addition of the 10kHz carrier does not alter discomfort compared with conventional pulses [1]. Did the authors attempt stimulation without the carrier frequency?

Which nerve roots were being targeted by the electrode placement. How was correct placement of electrodes, with respect to targeted muscles, verified?

Lines 77-78: It is unclear how informing parents/children of the stimulation intensity would bias the outcome when no control group was included in the trial.

Please report the stimulation intensities used, and the stimulation intensity at which motor threshold was reached, for all muscles measured. Were different intensities used for the electrode at C5-6 and T11-12? Was the stimulation split across the two cathode locations, or were they independent of one another?

Lines 78-79: Justify the use of sub-threshold stimulation. The current distribution is known to vary considerably with changes in posture; was the intensity modified according to posture, or how was this dealt with?

Lines 108-110: Please define the quantitative measures from treadmill and overground stepping. Was it quality of walking (if so, how was this assessed), walking distance/speed, or something else?

These outcome measures do not align with the objectives provided in the Introduction: no objective in relation to walking speed/quality was provided. How was “functional carry over to the home setting between sessions” (lines 54-55) assessed?

Line 111: Define PedsQL and provide a reference for its validity in this population.

Results:

Lines 113-114: was the data normally distributed?

Lines 118-119: Please provide a reference to support the normative ranges provided.

Where are results from the treadmill and overground stepping outcome measures presented?

Discussion:

First paragraph: I feel some of the information provided here is background, and would fit better in the introduction than discussion. The effects of other interventions on GMFM88 scores is clearly an important comparison for the discussion, but the limitations of interventions such as dorsal root rhizotomy, peripheral electrical stimulation, strengthening, “standard of care”, strategies to “avoid spasticity” need to be introduced in the background section rather than in the discussion.

In addition, the authors are not clear about the separation of ABNT and spinal stimulation. They describe their therapy as “spinal neuromodulation” and it is unclear whether this term relates to the combination of ABNT+spinal stim or to one of these interventions specifically. Has ABNT alone been trialled in this patient group before? If so, please present the findings from those studies. It is extremely difficult, from this trial alone, to separate the effects of ABNT (which we know can improve motor recovery by itself) from those of spinal stimulation alone and ABNT+spinal stim. The authors need to provide a clearer separation of the possible effects of the two interventions, as well as their combined effects, and acknowledge that their study does not provide sufficient evidence that these effects would not have been seen from ABNT alone.

Line 179: please define exactly what the “specific combination of motor pools” was, and how the authors verified that they were being reached in each subject.

Line 187: please define what “currently prescribed standard of care” is (in the introduction).

Line 204: As above, the authors need to provide more details on how they set stimulation intensity, and how much “below motor threshold” it was.

Lines 205-206 and 209-212: the authors must provide evidence for these broad statements or acknowledge that they are hypothetical. They have not presented any EMG or neurophysiology in their results to support this concept in their subjects. I would suggest the authors remove this statement or provide evidence from studies that have used neurophysiological measures to support it. I don't believe the mechanisms behind spinal stimulation are yet understood, and this part of the discussion is too speculative.

Lines 213-216: As the authors did not measure the effects of spinal neuromodulation alone, their own data does not support this statement. Please provide evidence to support it or remove it.

Lines 217-220: This statement should be in the introduction. Does “standard of care” meet this criteria? How is ANBT different to this?

Lines 225-227: I cannot see how the present results demonstrate that spinal neuromodulation suppresses “the physiological state of ... aberrant connections”. As I have stated previously, no neurophysiology is reported to support this statement, and no quantification of body alignment with/without spinal neuromodulation was made. The observation that the patients moved “more normally” is subjective, and insufficient to support the claims being made.

Lines 229-231: “seem to have shown greater effectiveness when motivated by using a self-selected activity” In comparison to what? Again, there is insufficient evidence to support this.

Lines 237-239: The findings do not support this statement. Please replace it with the findings of the study: the spinal stimulation + ABNT improved GMFM88 scores and PedsQL scores.

Figure 1B: define the blue/orange lines in this plot.

Figure 1C: Provide a description of the lines provided in the plot. Are they regression lines? If so, please provide R2 values and include a description of this analysis in the methods.

Minor points

Line 48: there is a typo in the reference (23).

Line 180: "principle" change to "principal"

Line 202: there is a typo

References

1. Manson, G.A., J.S. Calvert, J. Ling, B. Tychon, A. Ali, and D.G. Sayenko, The relationship between maximum tolerance and motor activation during transcutaneous spinal stimulation is unaffected by the carrier frequency or vibration. *Physiological reports*, 2020. 8(5): p. e14397.

Reviewer #2 (Remarks to the Author):

Summary of Key Findings:

Researchers found that 16 sessions across 8 weeks of transcutaneous electrical spinal cord stimulation in concert with 'activity-based neurorehabilitation therapy' significantly improved motor abilities in children with mild cerebral palsy (GMFCS I-II) and severe cerebral palsy (GMFCS IV-V) assessed with the GMFM-88.

The findings are a significant contribution to the field and especially to better understanding therapeutic interventions to improve motor abilities in children with CP. This builds on preliminary work (2017, 2021) in the CP population. In the Discussion, the authors add relevant literature and report and compare to other therapies and outcomes, i.e., GMFM88, for children with CP receiving other types of interventions.

Would it be possible with these changes that any participants' changed their GMFCS classification?

Again, comparing the current study and specific domain changes (currently not included) to other interventions and specific domain changes relative to the GMFCS LEVEL in the literature would add to the relevance of these specific findings and their potential value and uniqueness as a study of efficacy. The Discussion does address several studies in comparison; more detail may be valuable.

Abstract:

Line 12: The meaning of 'chronic effects' is unclear in this sentence. There are no follow-up or durability assessments of the findings after 16 sessions, i.e. no 'chronic' effects were measured. Please clarify.

Line 12: The particular device used to deliver stimulation may not need identification in the abstract but in the Methods section.

Line 14: Suggest instead of reporting 'ranging' from I to V, as Level III GMFCS level is not represented, that the authors report the exact 'n' of each, Level I-1, Level II-4, Level IV-3, Level V-6.

Line 18: 'transform the neural networks'....this refers to a potential mechanism for behavioral change. What does 'transform' mean specifically and was the 'transformation' measured.

Line 18: If you are referring to the outcomes of improved 'voluntary postural and locomotor activity', then specific outcomes need to address these two motor abilities. For instance, severely impaired postural control is consistent with those classified in GMFCS-Level V. What instrument demonstrated

improved voluntary postural control?

The Segmental Assessment of Trunk Control is a key instrument to assess trunk control in children who cannot independently sit for assessments and an excellent tool for persons classified as GMFCS V. Please indicate the measure used to assess 'voluntary postural control' and then the specific scores for this finding for each child.

How was 'locomotor activity' assessed? The Methods indicate 'stepping on a treadmill (with minimal support and assistance, as needed) and 'stepping overground (with minimal support and assistance, as needed). Are these assessments of independence? Of quality of step kinematics? Of being able to coordinate locomotion? Recommend clarification for potential reproducibility.

Line 19: The frequency and duration of treatment should be fully represented in the abstract, i.e., spinal stimulation and ABNT were delivered 2x/week for 8 consecutive weeks (total 16 sessions). Spell out GMFM88 in the abstract, first time it is used.

Introduction

Line 44: The use of the word 'recovery' is not suitable for children with CP who never had the typical neuromuscular capacity as born with dysfunctional pathways and experience has not been with typical movement patterns. Perhaps 'learning anew' is the direction or habilitation: defined as the assisting of a child with achieving developmental skills when impairments have caused delaying or blocking of initial acquisition of the skills.

Line 53: In reviewing the literature for CP, there appears to be one other key article published in 2017 addressing locomotor performance and use of neuromodulation and activity-based therapy in children with CP, prior to the 2021, Gad et al. reference.

Solopova IA, Sukhotina IA, Zhvansky DS, Ikoeva GA, Vissarionov SV, Baidurashvili AG, Edgerton VR, Gerasimenko YP, Moshonkina TR. Effects of spinal cord stimulation on motor functions in children with cerebral palsy. *Neurosci Lett.* 2017 Feb 3;639:192-198. doi: 10.1016/j.neulet.2017.01.003. Epub 2017 Jan 4. PMID: 28063935.

Line 54-54: Be sure that each outcome measure and the methodology for collection of data is provided in the Methods section. The 'functional carry over to the home setting between sessions' and how this information was acquired was not provided.

Methods:

Line 60: notes that 15 patients were recruited for the study. Did one drop out? Or otherwise? 14 patients are reported in the abstract and in the methods section table.

Line 64-68: Many children with abilities classified as GMFCS Level IV and V are at risk for developing scoliosis. Was this a consideration for exclusion? If not, did any of the participants have scoliosis and to what degree? Is this a factor for consideration and what impact might it have on conducting transcutaneous electrical spinal stimulation?

Table 1. Additional information concerning the CP diagnosis would be helpful for interpretation of the data, e.g., spastic diplegia, hemiplegia (R/L), spastic quadriplegia, athetoid. Also, are GMFMCS levels reported as 'numbers' or Roman numerals?

Line 72: Reference to a therapy based on the brand name of a device is inconsistent with naming a therapy based upon the goal/intent and then identifying what equipment is used, e.g., strengthening via theraband vs. theraband therapy, or locomotor training via a robotic device or manually-assisted cues vs. Lokomat therapy. The device is the instrument for delivery and not the therapy itself. How the device is used matters; how clinical decisions are made matters. The intent is to provide transcutaneous electrical spinal stimulation; the SCONE is the particular device that was used to

deliver this therapy and important, but not the therapy itself.

Thus, recommend not titling this section, SCONE Therapy but instead Transcutaneous Electrical Spinal Cord Neuromodulation or Stimulation.

Description of the transcutaneous electrical spinal stimulation intervention/delivery would benefit from greater detail for future reproducibility.

For instance, what were the size of the electrodes? Did this vary according to size/age of the participant? How were the intensities selected? The method of 'sub-threshold' application is made, but the methods lack a description of what particular 'motor' threshold was established and how and what specific muscles were assessed for muscle contractions. As this may be somewhat different compared to studies of children/adults with spinal cord injury, a clear explanation/description would be valuable for others pursuing this line of inquiry or even for possible clinical use in the future.

What were the specific intensities of stimulation at each site? Did these change across time/sessions? Was the stimulation intensity re-evaluated across the sessions? Report the stimulation intensities for each participant and each site or at minimum the mean, SD, range.

ABNT: Much of the ABNT is described as functional activities. Why were these activities selected?

Can you provide references for ABNT as used in this study and delineate the difference in practicing daily, functional activities and ABNT?

The term, activity-based therapies, has been used in the literature and it still requires standardization and clear descriptions whenever it is used as an intervention. Several articles in the human literature provide some background and even differences that may compare/contrast with the ABNT/intervention delivered.

Roy, R.R., Harkema, S.J., and Edgerton, V.R. (2012). Basic concepts of activity-based interventions for improved recovery of motor function after spinal cord injury. *Arch Phys Med Rehabil* 93(9), 1487-1497. doi: 10.1016/j.apmr.2012.04.034.

Sadowsky, C.L., and McDonald, J.W. (2009). Activity-based restorative therapies: concepts and applications in spinal cord injury-related neurorehabilitation. *Dev Disabil Res Rev* 15(2), 112-116. doi: 10.1002/ddrr.61.

Dolbow, D.R., Gorgey, A.S., Recio, A.C., Stiens, S.A., Curry, A.C., Sadowsky, C.L., et al. (2015). Activity-Based Restorative Therapies after Spinal Cord Injury: Inter-institutional conceptions and perceptions. *Aging Dis* 6(4), 254-261. doi: 10.14336/AD.2014.1105.

Behrman AL, Ardolino EM, Harkema SJ. Activity-Based Therapy: From Basic Science to Clinical Application for Recovery After Spinal Cord Injury. *J Neurol Phys Ther.* 2017 Jul;41 Suppl 3(Suppl 3 IV STEP Spec Iss):S39-S45. doi: 10.1097/NPT.000000000000184. PMID: 28628595; PMCID: PMC5477660.

Did children also remove any braces during this period or were they allowed to wear braces during the study, whether during the experimental intervention or at home? Why were aqua- and hippotherapy specifically allowed and other therapies disallowed?

Further clarify differences in the ABNT for participants Level I-II and IV-V.

Outcome measures:

The GMFM88 should be described as to what it entails, psychometric properties, validity, reliability in CP population and thus selected among outcome measures to assess what specifically.

Recommend reporting individual domains and outcomes of the GMFM88 for this study and its participants. The variability in the initial abilities across participants likely varies across Levels represented and individuals. Knowing what abilities particularly improved is of value and across levels.

'Stepping on a treadmill' and 'stepping overground' needs specifics to understand what is being assessed and how? Descriptive, observational, kinematics, EMG?

References should be provided in the methods section.

Also, who conducted the assessments (specific training) and were the assessors blinded to the study, its purpose, or otherwise? If so, please indicate, if not, please indicate. Did they conduct both pre- and post-assessments, live or videotaped?

Results:

Indicate the adherence of participants to the schedule of interventions, 2x/week for 16 sessions for 8 consecutive weeks. Thus, if absences, were these made up at another time? So, everyone completed 16 sessions? What other training data can be provided: time spend in each activity? The specific activities?

Appreciate the inclusion of no adverse events particularly working with pediatric population. Is there a temporary red/pink under the stimulation electrodes that dissipates?

Line 118-119. Add references.

Line 120-121. Need data in results to justify this statement. What is the immediate improvement observed or scored? What was/were the positive outcomes and what percentage of participants, Levels, etc.

Line 121, More specifically, 'after completing the 16 sessions of stimulation combined with ABNT' GMFM88 scores improved. Is there any interim data, e.g., after 8 sessions to indicate the rate of change/improvement? What domains improved?

Line 123, $P < 0.05$

Line 127. Was there an analysis completed to assess impact of age on outcomes? Otherwise, perhaps simply an observation, but without analysis.

Line 131-132, Supplementary videos.

Greater explanation is necessary with the videos.

What was being assessed pre-intervention and post-intervention for stepping on a treadmill. Shoes were worn in one video at the start and not the post-training video as child is barefoot. This could have a difference in response to the treadmill and load. Is there a way to quantify what was observed?

Stable assistance was provided at the trunk and pelvis that appears to be quite static and not providing dynamic movements of the pelvis/trunk consistent with stepping. The treadmill speed and 'load' (unknown) were then the two afferent inputs. Identifying the treadmill speed and the load would be valuable as well relative to delivery of this aspect of the activity-based 'neurorehabilitation' intervention?

Using the term 'kid' in the video title is quite informal. Leave this to the Editors, but prefer term 'child' or 'participant' in a scientific journal.

Line 134. Who observed or reported these findings? It is a broad, sweeping statement without explanation of how observed, when, by who, how many times, assessment. Provide detailed description of these outcomes, e.g., parent interview, observed by clinician at what timepoints, recorded by what method, video?

Line 138-149. Again, who observed these? What is 'more relaxed'? Smiled more frequently....not sure how to interpret this 'observation'.

Line 140. More detailed information of the Peds QoL would be valuable to understand instead of a score change. What specifically changed for family members? Did any of the participants complete the QoL assessment? Did this complement any anecdotal findings?

Discussion

Line 185, Suggest listing 16 sessions (if all achieved) within 8 weeks instead of 8 weeks of intervention.

Appreciate the robust finding relative to the heterogeneity of the population. Understanding specific presentations would further this point, e.g., hemiplegia, spastic dyplegia.

Line 260-220. Is this statement consistent with the description of the selection of functional activities in the methods section?

Can you address development across domains? Does potential change in one domain, locomotor, impact another? Developmental theories may support such findings, i.e., Adolph and Koch, 2019.

Line 239 'near normal postural and locomotor functions'. This statement may be a leap. Specific testing of these two capacities even beyond the GMFM88 may be important to make such a very strong interpretation of the current data.

In addition, 'normal' function would also mean longevity and durability, repeatability.

The Peds QoL outcome is not discussed. What is gained from this outcome?

Recommend a Limitations section, e.g., durability not assessed, blinded assessors (?), skewed GMFCS levels and III not represented, what else?

What next?

Reviewer #3 (Remarks to the Author):

This is a research report of 14 children with cerebral palsy who have undergone a combination of activity-based neurorehabilitation with transcutaneous electrical spinal cord stimulation over the course of 8 weeks. The subjects were categorized using the gross motor function classification scale as ranging from level I to V. Participants were tested using the gross motor function measure-88 as the primary outcome measure both before treatment and after the 8 weeks of treatment. The abstract mentions assessing treadmill stepping and overground stepping however those results were not quantified nor reported. Anecdotal reports of improvements in other functions are mentioned as are the results of improvement in quality of life (the PedQoL survey). The authors report that all subjects responded to treatment and improved in their GMFM88 scores. It is encouraging to see that electrical stimulation is being utilized in these populations and I am hopeful for this technology. However, I have concerns outlined below:

1) It's not clear that both the activity-based training plus the transcutaneous stimulation are needed for the improvements reported here. In fact several studies show that activity-based training or goal-directed physical activity alone can lead to improvements in motor function in people with cerebral palsy (Larsen et al., 2021; Mirich et al., 2021; Zoccolillo et al., 2016; Valentin-Gudiol et al., 2017; Lauruschkus et al., 2017). Were there any subjects which received only physical training or only spinal stimulation? It's difficult to draw conclusions that the transcutaneous stimulation is necessary or even helpful in this context.

2) Who performed the GMFM88 scoring? Was it blinded or done with an outside clinician for an unbiased assessment?

REVIEWER COMMENTS

Reviewer #1 (Remarks to the Author):

This manuscript explores an intervention that combines activity-based neurorehabilitation therapy (ABNT) with transcutaneous spinal stimulation on gross motor function in children with CP. Children with CP were recruited to undertake an 8-week intervention of spinal stimulation combined with ABNT. Before and after the intervention, outcome measures (gross motor function scale and a pediatric quality of life questionnaire) were conducted.

Interest in the application of transcutaneous spinal stimulation following spinal cord injury has gained considerable momentum in the past few years. The application to children with CP is relatively novel and interesting. The authors have conducted an important clinical trial, worthy of publication, however there are several major issues in the manuscript that need to be addressed prior to publication. The main issue is that the authors make broad and categorical statements and claims throughout the manuscript, with insufficient evidence (either from their own results or the literature) to support them. In particular, there are several claims throughout the discussion and conclusion that are not supported by the results of the study. The discussion and conclusion need to be rewritten with a clear distinction between the findings of the study and speculation about the mechanisms behind the results, which the authors have not attempted to measure in the present study.

Response: We thank the reviewer for the positive feedback and hope the revised manuscript addresses their concerns.

In the introduction, the authors provide insufficient background on standard of care, and the limitations of current interventions. In particular, the authors need to provide a more detailed description of ABNT, if and how this differs from standard of care, and evidence to support the effectiveness of ABNT given alone. Indeed, throughout the manuscript, the authors do not adequately address that they are exploring a combination of two interventions, with no control group, so it is not possible for them to separate the effects of each intervention, nor to say whether similar results would have been achieved with ABNT or spinal stimulation given alone.

Response: The reviewer is correct in pointing out that our intervention consists of two components, the ABNT in presence of spinal neuromodulation. The lack of control group is addressed in the limitations sections of the discussion. We have provided more details regarding the methods used in ABNT and also added further details regarding the current standard of care (SoC), improvements observed with SoC and their limitations. The text now reads “Pediatric Physical Therapy Standard of Care (SoC) for treatment for CP often include one or more of the following: 1) an activity-based neurorehabilitation therapy (ABNT) play-based approach, appropriate for age, with therapist facilitating more normal movement patterns with the goal of strengthening selected groups of muscles; 2) direct orthopedic strengthening exercises, often used in combination with stretching to maintain range of motion, and 3) emphasizing movement for sustaining metabolic and cardiovascular fitness and maintaining the ability of the child to move and function to his/her best ability, and become more independent through utilizing devices to help them sit, stand, or walk. Historically, the change in GMFM88 scores observed for currently used interventions ranges from 1.3 to 13.4 points over the course of 3 weeks to 18 months of treatment^{8,9}. More importantly, while being effective in improving GMFM88 scores, these interventions seem to have little effect in acquiring an expanded range of self-initiated and well-controlled movements.

And

Activity based neurorehabilitation therapy (ABNT): The children were asked to take off their shoes and braces (if any) before each therapy session. Each therapy session began with 1) treadmill training to provide maximum proprioceptive input to prime the nervous system. The treadmill speeds ranged from 0.1m/s to 0.5m/s based on the child's capability. Assistance was provided as needed by therapist(s) including i) moving limbs while stepping with the goal being to accurately place the heel and toes on the treadmill belt and achieve consistency of gait, ii) appropriate weight-shifting at pelvis and to advance their lower extremity during swing out in front of the Center of Mass and iii) maintain the head, shoulders, hips and heels in alignment. Next, ABNT included functional activities during play activities and walking with body support for alignment (as needed) including, 2) upright sitting, with trunk control and weight shifting during reaching, 3) transitions including sit to stand, rolling prone to a quadruped position, consecutive rolling etc., 4) dynamic standing with postural and weight shifts, 5) side-stepping with frontal plane weight shifts, 6) backward stepping, 7) Climbing/creeping up an incline, 8) climbing up and down steps and 9) over ground walking. Children classified as levels I and II underwent all the components of the therapy listed above, whereas those classified as levels IV and V focused more on activities 1-6. All children were trained 2 sessions a week for 8 weeks with each session lasting for 50-60 mins. In case a child missed a session, a make-up session was performed within 7 days to ensure completion of 16 sessions within 8 weeks. During all therapeutic activities, it was ensured that appropriate posture, midline orientation and head position was maintained with maximal weight-bearing to optimize proprioceptive information being processed by the nervous system. Since children with GMFCS levels I and II are functional, but with limited motor patterns available, the objective was to enable self-generated movement patterns with more typical posture and alignment, whereas lesser functioning children with GMFCS levels IV and V, the objective was to allow for new motor patterns to evolve during play activities, while retraining existing patterns by using proper posture and alignment. Toys with lights and sounds were used to engage children to facilitate movements including reaching, grasping and visual tracking etc. All children were asked to discontinue other therapies, except for "aqua"- and "hippotherapy", prior to initiation of and during the study²⁴. Both aqua and hippo therapies are typical activities for children, and they do not place excessive motor demands on the child.

The objectives stated at the end of the introduction are not met by the outcome measures, and the results of all outcome measures are not presented in the results.

Response: We have clarified the objectives in the introduction and outcome measures in the methods that are consistent with our results. The text now reads "We hypothesized that, spinal neuromodulation and appropriate proprioception during ABNT, can facilitate learning of sensorimotor function in children with CP. The changes in gross motor function over time for children with CP is typically measured with the Gross Motor Function Measure (GMFM) scale. Thus, the objective of our study was to determine the effect of ABNT in the presence of spinal neuromodulation in children with CP on the GMFM88 scale. "

One major point is that substantially more detail is required about the stimulation parameters used, how they were determined and their justification (see points below under methods).

Response: We have added the intensities ranges to the method section. The text now reads "The intensities over the C5-6 spine ranged between 12 to 18mA and over the T11-12 ranged between 10 to 16mA based on the activity being performed. The initial amplitude for each child was set when the child attempted to extend the cervical or thoracic regions. During activities involving sitting, rolling etc. amplitudes were lowered by 1-2mA, whereas, during standing and stepping, the intensities were

increased by 2-4mA. During the course of a given activity, the intensities would be modulated ± 2 mA based on observed functional performance of the child. ”.

Please modify the Title to better reflect the results presented in the manuscript.

Response: We have modified the title to reflect the results. The new title reads “Noninvasive spinal neuromodulation used concurrently with activity-based neurorehabilitation improves voluntary sensorimotor function in children with cerebral palsy”

Abstract

Lines 9-11: there is insufficient evidence to support the statement “Spinal neuromodulation and activity-based rehabilitation triggers neural network reorganization”. Please modify.

Response: We have modified the statement.

Lines 17-18: the results presented in the paper do not support the statement “We demonstrate that transcutaneous spinal neuromodulation during ABNT can transform the neural networks in children diagnosed with CP to improve voluntary postural and locomotor activity”. No neurophysiology or imaging was done to explore whether changes in neural networks occurred. In addition, no control group was included. Please replace this statement with the actual results that are presented in the paper.

Response: We have modified the statement and it now reads, “We demonstrate that the combination of transcutaneous spinal neuromodulation and ABNT resulted in clinically and statistically significant improvement of voluntary sensorimotor function in children diagnosed with CP based on the change in GMFM88 scores within 8 weeks of treatment.

We disagree, however, with the Reviewer’s viewpoint which assumes that reorganization of networks can be demonstrated by only neurophysiology and imaging, when in fact neither is a more direct demonstration than the changes in EMG patterns which is a direct functional demonstration in *in vivo* conditions, not a “biomarker” of reorganization. We have demonstrated these results in our previous study which is the basis for the current study.

Introduction:

Lines 47-48: “humans that have lost proprioception as an adult are functionally paralyzed, even though all descending motor pathways remain intact^{22,23}” The references provided do not support the statement. The studies by Gordon et al. and Ghez et al. (1995) present findings on people with proprioceptive deficits due to large-fibre sensory neuropathies. They report that that the patients made large spatial errors in movement, compared with controls, and that their accuracy improved with visual feedback. The studies do not suggest that these patients were unable to move or “functionally paralyzed”. I would suggest the authors remove the statement unless they can provide appropriate references to support it.

Response: There are limited references that describe this unique situation wherein proprioception is lost without any other neurological disorders conditions. Based on the description of this particular case, the individual was unable to control his limbs voluntarily in the absence of visual feedback, i.e., being functionally paralyzed. Only after months of practice could he even sit-up and with practice stepping, he could do so with a robotic-like gait with careful visualization

Lines 48-49: “spinal networks can readily learn to perform a new motor task without input from the

brain^{24,25} Please clarify that this is true in Mammals but not in humans, as evidenced by several Locomotor Treadmill Training trials in humans with SCI.

Response: We have deleted the statement.

Lines 49-50: “neuro plasticity can occur in spinal and supraspinal networks with spinal neuromodulation combined with ABNT^{26,27}” Again, the statement is not supported by the references. These studies show that neuromodulation of lumbosacral networks enables specific functional movements, but provide no direct evidence that neuroplasticity has taken place. In addition, both of these studies were conducted using implanted (epidural) electrodes, with no kHz carrier frequency, which is rather different to the neuromodulation being applied in the present study.

Response: Our experience working with individuals with spinal cord injuries using transcutaneous spinal neuromodulation have successfully demonstrated that significant levels of neuroplasticity is possible in the brain and spinal cord. We have added the appropriate references related to use of transcutaneous spinal neuromodulation.

Line 54: Please define GMFM88, and provide references to support its validation in children with CP.

Response: The definition of GMFM88 has been added to the introduction.

Methods:

Line 63: was there an upper age limit?

Response: No, we did not set an upper age limit

Table 1: Define GMFCS. Be consistent with decimal places.

Response: full form for GMFCS has been added.

Lines 74-77: Please provide justification for the parameters chosen. Recently, it has been reported that the addition of the 10kHz carrier does not alter discomfort compared with conventional pulses [1]. Did the authors attempt stimulation without the carrier frequency?

Response: As described in Manson et al., 2020, “In both the single-pulse and continuous stimulation protocols, participants tolerated significantly higher levels of stimulation with the carrier frequency paradigm compared to the other stimulation paradigms. However, when the maximum tolerable stimulation intensity of each stimulation paradigm was normalized to the intensity required to evoke a lower limb muscle response, there were no statistical differences between the stimulation paradigms.” Both the conditions tested were at significantly higher intensities of stimulations i.e., 200-500mA ranges. Whereas the intensities used in the current study were between 10-20mA. Finally, the objective of the study was not to compare presence and absence of the high frequency, rather to study the functional improvements possible.

Which nerve roots were being targeted by the electrode placement. How was correct placement of electrodes, with respect to targeted muscles, verified?

Response: As stated in the discussion and previously described, spinal neuromodulation is not meant to target a single set of nerves, rather a large neural network in the spinal cord that is responsible for control of movement. The recent paper from Courtine’s group emphasizes the necessity of a very specific location for electrode placement. There is abundant information reported that placement of a single electrode relative to a single dorsal root is of minor importance in identifying effective placement of multiple electrodes, particularly transcutaneously and when the target is multiple

lumbosacral segments as need for locomotor functions.

Lines 77-78: It is unclear how informing parents/children of the stimulation intensity would bias the outcome when no control group was included in the trial.

Response: Some of the individuals in the study were old enough to understand the stimulation parameters used. Thus, to maintain the same standard across all patients, the stimulation intensity was not revealed during the training sessions.

Please report the stimulation intensities used, and the stimulation intensity at which motor threshold was reached, for all muscles measured. Were different intensities used for the electrode at C5-6 and T11-12? Was the stimulation split across the two cathode locations, or were they independent of one another?

Response: The range of stimulation intensities has been added to the methods section. We have also added figure 1 that describes the location of electrodes and the ground electrodes.

Lines 78-79: Justify the use of sub-threshold stimulation. The current distribution is known to vary considerably with changes in posture; was the intensity modified according to posture, or how was this dealt with?

Response: The use of sub-motor threshold intensities has been developed by our team and tested in rodents and humans since 2013 (Gad et al., 2013). The intensities were adjusted based on the ongoing visually assessed motor performance of the child while engaged in play activities, keeping in mind the response of potential changes in threshold, based on observed posture and need.

Lines 108-110: Please define the quantitative measures from treadmill and overground stepping. Was it quality of walking (if so, how was this assessed), walking distance/speed, or something else?

Response: The outcome related to walking was assessed qualitatively, i.e., being able to take weight bearing steps independently, being able to take steps with some assistance (weight bearing and/or assistance to move limbs) and being able to take steps with full assistance (weight bearing and assistance to move limbs). We have added details to the methods and results sections. The text in the methods reads "At the start of therapy, 9 children were non-ambulatory and needed assistance while stepping whereas 6 were able to take some steps. At the end of 8 weeks of therapy, only 4 children were non ambulatory, 5 were capable to stepping with minimal assistance and 6 were able to step independently with no external assistance (Fig. 3D and 3F)."

These outcome measures do not align with the objectives provided in the Introduction: no objective in relation to walking speed/quality was provided. How was "functional carry over to the home setting between sessions" (lines 54-55) assessed?

Response: We have modified the introduction accordingly. We have now stated the change in the GMFM88 scores as the objective of the study.

Line 111: Define PedsQL and provide a reference for its validity in this population.

Response: The appropriate reference has been added.

Results:

Lines 113-114: was the data normally distributed?

Response: The data were not tested for normality, however, the observed scores are in line with what has been reported in literature.

Lines 118-119: Please provide a reference to support the normative ranges provided.

Response: The ranges for GMFM88 for the four levels tested and corresponding reference has been included. The text now reads “All children demonstrated GMFM88 scores consistent for their age and GMFCS level at the start of the therapy period with scores ranging from 68 to 85 for levels I and II and 2.9 to 29 for levels IV and V (historically reported scores: level I, 76.29 ± 17.14 , level II, 59.51 ± 13.49 , level IV 30.21 ± 10.79 , and level 8.33 ± 5.16).”

Where are results from the treadmill and overground stepping outcome measures presented?

Response: The results currently read, “At the start of therapy, 9 patients were non-ambulatory and needed assistance while stepping whereas 5 were able to take some steps. At the end of 8 weeks of therapy, only 4 patients were non ambulatory, 5 were capable to stepping with minimal assistance and 5 were able to step independently with no external assistance.” Please see supplementary video 1 and 2.

Discussion:

First paragraph: I feel some of the information provided here is background, and would fit better in the introduction than discussion. The effects of other interventions on GMFM88 scores is clearly an important comparison for the discussion, but the limitations of interventions such as dorsal root rhizotomy, peripheral electrical stimulation, strengthening, “standard of care”, strategies to “avoid spasticity” need to be introduced in the background section rather than in the discussion.

Response: We have moved the relevant information in the introduction.

In addition, the authors are not clear about the separation of ABNT and spinal stimulation. They describe their therapy as “spinal neuromodulation” and it is unclear whether this term relates to the combination of ABNT+spinal stim or to one of these interventions specifically. Has ABNT alone been trialled in this patient group before? If so, please present the findings from those studies. It is extremely difficult, from this trial alone, to separate the effects of ABNT (which we know can improve motor recovery by itself) from those of spinal stimulation alone and ABNT+spinal stim. The authors need to provide a clearer separation of the possible effects of the two interventions, as well as their combined effects, and acknowledge that their study does not provide sufficient evidence that these effects would not have been seen from ABNT alone.

Response: The objective of the study was not to study the differences between ABNT and spinal neuromodulation, rather to study the new model we are developing which comprises of the combination of the two.

The current standard of care (SoC), i.e., PT/OT has shown limited success in improving functional movement. The improvements in GMFM88 scores ranged from 1.3 to 6.2 points after 3 weeks to 8 months of SoC PT (Bakanjee et al., 2018). Further, none of the current SoC PT methods have demonstrated improvement in voluntary initiation of sensorimotor function. In addition, our own studies in individuals with SCI has demonstrated minimal change in function and voluntary capabilities with intense PT only until the introduction of spinal neuromodulation. Thus, these previous studies highlight the minimal change possible and inability to initiate voluntary movements without spinal neuromodulation.

Line 179: please define exactly what the “specific combination of motor pools” was, and how the authors verified that they were being reached in each subject.

Response: This approach stems from and leverages years of experience working with adults with spinal cord injuries, numerous rodent and non-human primate mapping studies and recent acute studies in children with CP that demonstrates the regions of the spinal cord that contain the intrinsic circuits responsible for control of movement in children with CP. Furthermore, the motor performance is defined more by the coordination patterns of the motor pools rather than which pools are involved. Basically, for example virtually the same motor pools are activated when stepping forward or backward, but the pattern of activation defines the movement.

Line 187: please define what “currently prescribed standard of care” is (in the introduction).

Response: We have added a paragraph that describes the current standard of care. We have modified the sentence to reference the previous studies. It now reads.

Lines 213-216: As the authors did not measure the effects of spinal neuromodulation alone, their own data does not support this statement. Please provide evidence to support it or remove it.

Response: We have removed the sentence.

Lines 217-220: This statement should be in the introduction. Does “standard of care” meet this criteria? How is ANBT different to this?

Response: We have moved this sentence to the introduction. The activities covered within ABNT are in line with this guideline.

Lines 225-227: I cannot see how the present results demonstrate that spinal neuromodulation suppresses “the physiological state of ... aberrant connections”. As I have stated previously, no neurophysiology is reported to support this statement, and no quantification of body alignment with/without spinal neuromodulation was made. The observation that the patients moved “more normally” is subjective, and insufficient to support the claims being made.

Response: This statement is hypothetical based on our previous acute study in children with CP using spinal neuromodulation and multiple acute and chronic studies in adults with SCI. We have modified it to clarify our position and added the corresponding references to support this statement. The text now reads “**Based on our previously demonstrated neurophysiological results²⁶, we hypothesized that the physiological state of these aberrant connections can be suppressed with spinal neuromodulation to a level that allows the children to voluntarily initiate the intended movement, more normally, and thus generate more normal patterns of proprioception derived from improved signals generated by the spinal networks.**”

Lines 229-231: “seem to have shown greater effectiveness when motivated by using a self-selected activity” In comparison to what? Again, there is insufficient evidence to support this.

Response: As currently stated, “seem to have shown greater effectiveness when motivated by using a self-selected activity (in pediatrics usually play related) rather than requiring a more standard, but less interesting activity for the child.” This strategy and observation is in line with the guidance from WHO and CPA and based on our teams experience working with children with CP, and is supported in the previous motor learning studies (Kleim JA et al., 2008)

Lines 237-239: The findings do not support this statement. Please replace it with the findings of the study: the spinal stimulation + ABNT improved GMFM88 scores and PedsQL scores.

Response: We have modified the statement as recommended.

Figure 1B: define the blue/orange lines in this plot.

Response: The blue and orange legends are added.

Figure 1C: Provide a description of the lines provided in the plot. Are they regression lines? If so, please provide R2 values and include a description of this analysis in the methods.

Response: The R2 values are added for the regression lines.

Minor points

Line 48: there is a typo in the reference (23).

Response: The typo has been edited.

Line 180: "principle" change to "principal"

Response: The typo has been edited.

Line 202: there is a typo

Response: The typo has been edited.

References

1. Manson, G.A., J.S. Calvert, J. Ling, B. Tychhon, A. Ali, and D.G. Sayenko, The relationship between maximum tolerance and motor activation during transcutaneous spinal stimulation is unaffected by the carrier frequency or vibration. *Physiological reports*, 2020. 8(5): p. e14397.

Reviewer #2 (Remarks to the Author):

Summary of Key Findings:

Researchers found that 16 sessions across 8 weeks of transcutaneous electrical spinal cord stimulation in concert with 'activity-based neurorehabilitation therapy' significantly improved motor abilities in children with mild cerebral palsy (GMFCS I-II) and severe cerebral palsy (GMFCS IV-V) assessed with the GMFM-88.

The findings are a significant contribution to the field and especially to better understanding therapeutic interventions to improve motor abilities in children with CP. This builds on preliminary work (2017, 2021) in the CP population. In the Discussion, the authors add relevant literature and report and compare to other therapies and outcomes, i.e., GMFM88, for children with CP receiving other types of interventions.

Would it be possible with these changes that any participants' changed their GMFCS classification?

Response: We believe with longer term ABNT and spinal neuromodulation, there may be a change in the GMFCS levels. However, this is beyond the scope of the current study and will be addressed in future studies.

Again, comparing the current study and specific domain changes (currently not included) to other interventions and specific domain changes relative to the GMFCS LEVEL in the literature would add to the relevance of these specific findings and their potential value and uniqueness as a study of efficacy. The Discussion does address several studies in comparison; more detail may be valuable.

Response: We have added the comparison for specific therapies and the corresponding changes in GMFM88 scores and compared them to our results. However, since the MCID has only been defined for the total score, we are limiting the comparison to the total score only

Abstract:

Line 12: The meaning of 'chronic effects' is unclear in this sentence. There are no follow-up or durability assessments of the findings after 16 sessions, i.e. no 'chronic' effects were measured. Please clarify.

Response: We have deleted the word chronic.

Line 12: The particular device used to deliver stimulation may not need identification in the abstract but in the Methods section.

Response: We have deleted the reference to the device from the abstract.

Line 14: Suggest instead of reporting 'ranging' from I to V, as Level III GMFCS level is not represented, that the authors report the exact 'n' of each, Level I-1, Level II-4, Level IV-3, Level V-6.

Response: We have updated the text accordingly.

Line 18: 'transform the neural networks'....this refers to a potential mechanism for behavioral change. What does 'transform' mean specifically and was the 'transformation' measured.

Response: The transformation is referring to the ability to learn to generate self-initiated voluntary movements after just 16 sessions of therapy.

Line 18: If you are referring to the outcomes of improved 'voluntary postural and locomotor activity', then specific outcomes need to address these two motor abilities. For instance, severely impaired postural control is consistent with those classified in GMFCS-Level V. What instrument demonstrated improved voluntary postural control?

Response: Voluntary effort is captured on the GMFM88 tests. Voluntarily refers to the fact that the children in Levels IV and V began to attempt to initiate movement when verbally requested to do so, which was impossible for them. The changes in GMFM88 scores also occurs since the children tried to do the activity, even though they were unsuccessful, whereas previously they would not attempt to move and were obviously unsuccessful.

The Segmental Assessment of Trunk Control is a key instrument to assess trunk control in children who cannot independently sit for assessments and an excellent tool for persons classified as GMFCS V. Please indicate the measure used to assess 'voluntary postural control' and then the specific scores for this finding for each child.

Response: The GMFM88 is the gold standard for measuring and assessing self-initiated voluntary sensorimotor function in children with CP.

We have added the different dimensions of the GMFM88 to the introduction that includes A) lying and rolling, B) sitting, C) crawling and kneeling, D) standing, E) walking, running and jumping. The statistically and clinically significant increase in GMFM88 scores represent the 'improved voluntary postural and locomotor activity' The Gross Motor Classification Scale (GMFCS) levels have descriptors describing the postures in assigning Levels of CP I-V. For instance, children with Level V cannot sit by themselves due to weakness and postural instability, requiring external equipment to do so.

How was 'locomotor activity' assessed? The Methods indicate 'stepping on a treadmill (with minimal

support and assistance, as needed) and 'stepping overground (with minimal support and assistance, as needed). Are these assessments of independence? Of quality of step kinematics? Of being able to coordinate locomotion? Recommend clarification for potential reproducibility.

Response: The locomotor activity assessments were done qualitatively. The results currently read “At the start of therapy, 9 patients were non-ambulatory and needed assistance while stepping whereas 5 were able to take some steps. At the end of 8 weeks of therapy, only 4 patients were non ambulatory, 5 were capable to stepping with minimal assistance and 5 were able to step independently with no external assistance.”

Line 19: The frequency and duration of treatment should be fully represented in the abstract, i.e., spinal stimulation and ABNT were delivered 2x/week for 8 consecutive weeks (total 16 sessions).

Spell out GMFM88 in the abstract, first time it is used.

Response: We have added this to the abstract.

Introduction

Line 44: The use of the word 'recovery' is not suitable for children with CP who never had the typical neuromuscular capacity as born with dysfunctional pathways and experience has not been with typical movement patterns. Perhaps 'learning anew' is the direction or habilitation: defined as the assisting of a child with achieving developmental skills when impairments have caused delaying or blocking of initial acquisition of the skills.

Response: We have modified it to ‘acquired’.

Line 53: In reviewing the literature for CP, there appears to be one other key article published in 2017 addressing locomotor performance and use of neuromodulation and activity-based therapy in children with CP, prior to the 2021, Gad et al. reference.

Solopova IA, Sukhotina IA, Zhvansky DS, Ikoeva GA, Vissarionov SV, Baidurashvili AG, Edgerton VR, Gerasimenko YP, Moshonkina TR. Effects of spinal cord stimulation on motor functions in children with cerebral palsy. *Neurosci Lett*. 2017 Feb 3;639:192-198. doi: 10.1016/j.neulet.2017.01.003. Epub 2017 Jan 4. PMID: 28063935.

Response: We have addressed this issue in our previous acute study (Gad et al., 2021). Solopova et al., 2017 focused on a combination of functional electrical stimulation combined with peripheral stimulation, and robotic training in improving motor function. This system differs completely from our approach of using ABNT and spinal neuromodulation for improved motor control in all movement. We have discussed this extensively in our previous manuscript

Line 54-54: Be sure that each outcome measure and the methodology for collection of data is provided in the Methods section. The 'functional carry over to the home setting between sessions' and how this information was acquired was not provided.

Response: As mentioned in the results, both parents and children self-reported improvements in function that carried over to their homes in between sessions. The text reads “In addition, during interviews with the study team, all parents reported that their child voluntarily began to practice the newly learned motor skills at home even in the absence of neuromodulation after two to four sessions, further enhancing neuroplasticity and increasing the child’s activity level. Some of these motor skills included transitioning from their bed to the floor and continuing to crawl (Fig. 3B), being

able to voluntarily turn (Fig. 3A) and slide off the couch and voluntarily dislodging things around the bed to grab the parents' attention. These new motor skills were sustained for several days in between two therapy sessions. Further, anecdotally, parents also reported that their children demonstrated improvement in many of the symptoms secondary to cerebral palsy including improvement in visual regard and eye tracking (Fig. 3E); decreased startled response to usual stimuli such as touch (Fig. 3C), disappearance of primitive reflexes, such as asymmetrical tonic neck reflex (Fig. 3G); organized and more normal sleep patterns including being able to sleep continuously through the night; being more attentive; and increased attempts at verbal communication (Fig. 3D and 3H). A consistent observation was that all children seemed more relaxed, smiled more frequently and were able to respond to verbal requests from parents (Fig. 3H). _”.

Methods:

Line 60: notes that 15 patients were recruited for the study. Did one drop out? Or otherwise? 14 patients are reported in the abstract and in the methods section table.

Response: The typo has been corrected. 16 patients were recruited, all the patients completed the study.

Line 64-68: Many children with abilities classified as GMFCS Level IV and V are at risk for developing scoliosis. Was this a consideration for exclusion? If not, did any of the participants have scoliosis and to what degree? Is this a factor for consideration and what impact might it have on conducting transcutaneous electrical spinal stimulation?

Response: Children with CP at all levels may exhibit scoliosis, but it is usually not “fixed scoliosis at younger ages and with proper positioning, the scoliosis is remains flexible or the angles are not sufficient to impact them functionally (breathing). Since this was our first in human chronic study in children with CP, scoliosis was not considered as a factor and none of the patients were diagnosed with scoliosis. This may be a factor to consider in future studies.

Table 1. Additional information concerning the CP diagnosis would be helpful for interpretation of the data, e.g., spastic diplegia, hemiplegia (R/L), spastic quadriplegia, athetoid.

Also, are GMFMCS levels reported as 'numbers' or Roman numerals?

Response: We have included the CP diagnosis to table 1 and modified GMFCS levels to roman numerical.

Line 72: Reference to a therapy based on the brand name of a device is inconsistent with naming a therapy based upon the goal/intent and then identifying what equipment is used, e.g., strengthening via theraband vs. theraband therapy, or locomotor training via a robotic device or manually-assisted cues vs. Lokomat therapy. The device is the instrument for delivery and not the therapy itself. How the device is used matters; how clinical decisions are made matters. The intent is to provide transcutaneous electrical spinal stimulation; the SCONE is the particular device that was used to deliver this therapy and important, but not the therapy itself.

Thus, recommend not titling this section, SCONE Therapy but instead Transcutaneous Electrical Spinal Cord Neuromodulation or Stimulation.

Response: We have modified the title to *Transcutaneous Spinal Neuromodulation Therapy*.

Description of the transcutaneous electrical spinal stimulation intervention/delivery would benefit from greater detail for future reproducibility.

For instance, what were the size of the electrodes? Did this vary according to size/age of the

participant? How were the intensities selected? The method of 'sub-threshold' application is made, but the methods lack a description of what particular 'motor' threshold was established and how and what specific muscles were assessed for muscle contractions. As this may be somewhat different compared to studies of children/adults with spinal cord injury, a clear explanation/description would be valuable for others pursuing this line of inquiry or even for possible clinical use in the future.

What were the specific intensities of stimulation at each site? Did these change across time/sessions? Was the stimulation intensity re-evaluated across the sessions? Report the stimulation intensities for each participant and each site or at minimum the mean, SD, range.

Response: We have added the details of the electrodes used, how the stimulation intensities. The text now reads "The intensities over the C5-6 spine ranged between 12 to 18mA and over the T11-12 ranged between 10 to 16mA based on the activity being performed. The initial amplitude for each child was set when the child attempted to extend the cervical or thoracic regions. During activities involving sitting, rolling etc. amplitudes were lowered by 1-2mA, whereas, during standing and stepping, the intensities were increased by 2-4mA. During the course of a given activity, the intensities would be modulated ± 2 mA based on observed functional performance of the child."

ABNT: Much of the ABNT is described as functional activities. Why were these activities selected? Can you provide references for ABNT as used in this study and delineate the difference in practicing daily, functional activities and ABNT?

Response: The ABNT activities used here are in line with the guidelines provided by Cerebral Palsy Alliance (CPA), World Health Organization (WHO) and based on our teams experience in treating children with CP. The updated references are added to the methods section.

The term, activity-based therapies, has been used in the literature and it still requires standardization and clear descriptions whenever it is used as an intervention. Several articles in the human literature provide some background and even differences that may compare/contrast with the ABNT/intervention delivered.

Roy, R.R., Harkema, S.J., and Edgerton, V.R. (2012). Basic concepts of activity-based interventions for improved recovery of motor function after spinal cord injury. *Arch Phys Med Rehabil* 93(9), 1487-1497. doi: 10.1016/j.apmr.2012.04.034.

Sadowsky, C.L., and McDonald, J.W. (2009). Activity-based restorative therapies: concepts and applications in spinal cord injury-related neurorehabilitation. *Dev Disabil Res Rev* 15(2), 112-116. doi: 10.1002/ddrr.61.

Dolbow, D.R., Gorgey, A.S., Recio, A.C., Stiens, S.A., Curry, A.C., Sadowsky, C.L., et al. (2015). Activity-Based Restorative Therapies after Spinal Cord Injury: Inter-institutional conceptions and perceptions. *Aging Dis* 6(4), 254-261. doi: 10.14336/AD.2014.1105.

Behrman AL, Ardolino EM, Harkema SJ. Activity-Based Therapy: From Basic Science to Clinical Application for Recovery After Spinal Cord Injury. *J Neurol Phys Ther.* 2017 Jul;41 Suppl 3(Suppl 3 IV STEP Spec Iss):S39-S45. doi: 10.1097/NPT.000000000000184. PMID: 28628595; PMCID: PMC5477660.

Response: The references mentioned above involve patients and concepts of spinal cord injury and focus primarily on locomotor training. Key differences that exist between the SCI model and the CP model is that in kids with CP, the injury occurs prior to development of the spinal neural networks whereas in SCI, the networks may be partially developed. In addition, while locomotor training is a

key component of ABNT, the ABNT activities needed for a child with CP are very different for the therapy that an adult or child with SCI may need. We have modified the text, it now reads “The children were asked to take off their shoes and braces (if any) before each therapy session. Each therapy session began with 1) treadmill training to provide maximum proprioceptive input to prime the nervous system. The treadmill speeds ranged from 0.1m/s to 0.5m/s based on the child’s capability. Assistance was provided as needed by therapist(s) including i) moving limbs while stepping with the goal being to accurately place the heel and toes on the treadmill belt and achieve consistency of gait, ii) appropriate weight-shifting at pelvis and to advance their lower extremity during swing out in front of the Center of Mass and iii) maintain the head, shoulders, hips and heels in alignment. Next, ABNT included functional activities during play activities and walking with body support for alignment (as needed) including, 2) upright sitting, with trunk control and weight shifting during reaching, 3) transitions including sit to stand, rolling prone to a quadruped position, consecutive rolling etc., 4) dynamic standing with postural and weight shifts, 5) side-stepping with frontal plane weight shifts, 6) backward stepping, 7) Climbing/creeping up an incline, 8) climbing up and down steps and 9) over ground walking.”.

Did children also remove any braces during this period or were they allowed to wear braces during the study, whether during the experimental intervention or at home? Why were aqua- and hippotherapy specifically allowed and other therapies disallowed?

Response: All children were asked to take off braces and shoes during the course of the training to maximize proprioceptive input as is compatible with our hypothesis. They used them at home as needed based on their and their parents experience. We have updated the methods to describe this. Only hippotherapy and aqua therapy were allowed to be continued since these are not weight bearing therapies and are typical activities many children, including typically developing) do regularly as leisure activities.

We have modified the text, it now reads “All children were asked to discontinue other therapies, except for “aqua”- and “hippotherapy”, prior to initiation of and during the study²⁴. Both aqua and hippo therapies are typical activities for children, and they do not place excessive motor demands on the child.”.

Further clarify differences in the ABNT for participants Level I-II and IV-V.

Response: We have provided the clarification on the activities done by levels I-II and IV and V.

Outcome measures:

The GMFM88 should be described as to what it entails, psychometric properties, validity, reliability in CP population and thus selected among outcome measures to assess what specifically.

Recommend reporting individual domains and outcomes of the GMFM88 for this study and its participants. The variability in the initial abilities across participants likely varies across Levels represented and individuals. Knowing what abilities particularly improved is of value and across levels.

Response: We have added details about the GMFM88 instrument in the introduction. While the 5 dimensions are assessed, the clinical changes are assessed based on the overall score, thus, the individual dimensions may not be as valuable as the overall score

'Stepping on a treadmill' and 'stepping overground' needs specifics to understand what is being assessed and how? Descriptive, observational, kinematics, EMG?

References should be provided in the methods section.

Response: As mentioned in the methods, the assessments done were qualitative in nature and did not consist of detailed EMG and kinematic analysis. We have presented the EMG and kinematic data in our earlier publication (Gad et al., 2021). See supplementary videos 1 and 2. The text has been updated, it now reads **“At the start of therapy, 9 children were non-ambulatory and needed assistance while stepping whereas 6 were able to take some steps. At the end of 8 weeks of therapy, only 4 children were non ambulatory, 5 were capable to stepping with minimal assistance and 6 were able to step independently with no external assistance (Fig. 3D and 3F).”**.

Also, who conducted the assessments (specific training) and were the assessors blinded to the study, its purpose, or otherwise? If so, please indicate, if not, please indicate. Did they conduct both pre- and post-assessments, live or videotaped?

Response: The assessments were done by an experienced Board Certified pediatric physical therapist, The same person was responsible for the pre and post assessment. All assessments were done live but were also video recorded for future references. The training was done by an extended team of therapist.

Results:

Indicate the adherence of participants to the schedule of interventions, 2x/week for 16 sessions for 8 consecutive weeks. Thus, if absences, were these made up at another time? So, everyone completed 16 sessions? What other training data can be provided: time spend in each activity? The specific activities?

Responses: The specific activities are listed in the methods section. We have added details regarding the kinds of activities performed based on the GMFCS levels.

Appreciate the inclusion of no adverse events particularly working with pediatric population. Is there a temporary red/pink under the stimulation electrodes that dissipates?

Response: Since the level of current used was relatively low (10-30mA), no redness was observed under the skin. There were no adverse events for any child at any time during the study.

Line 118-119. Add references.

Response: Reference has been added.

Line 120-121. Need data in results to justify this statement. What is the immediate improvement observed or scored? What was/were the positive outcomes and what percentage of participants, Levels, etc.

Response: We have updated the list the improvements observed. The text now reads **“Positive responses were observed in all children during the first ABNT therapy session with improvement in posture and sensorimotor capabilities consistent with previous results²⁶. For individuals with levels, I and II, improvements included the ability to take more independent steps with their center of mass (CoM) sufficient to sustain equilibrium, upright posture and improved balance. Children with level IV and V demonstrated more subtle improvements including better voluntary head control, ability to sit for longer durations without external support and lower assistance needed during stepping. Notably, all children became more interested in the chosen activity and often responded positively via facial expressions when successfully interacting with cause-and-effect toys. This was observed in almost all the children at Levels IV and V when previously, they did not seem interested in interacting with any toys.”**.

Line 121, More specifically, 'after completing the 16 sessions of stimulation combined with ABNT'

GMFM88 scores improved. Is there any interim data, e.g., after 8 sessions to indicate the rate of change/improvement? What domains improved?

Line 123, $P < 0.05$

Response: Patients were only tested at 8 weeks. Since the MCID is defined for the overall instrument, scores are only being compared for the total GMFM88 scores. However, parents have reported continued improvement in motor skills after taking part in the stimulation sessions. Posture and initiation of movement appear to persist, and even improve, after the study with many of the children.

Line 127. Was there an analysis completed to assess impact of age on outcomes? Otherwise, perhaps simply an observation, but without analysis.

Response: We have addressed the issue of age and GMFCS level in figure 1. While all the patients reported an increase in GMFM88 scores greater than the MCID, there seems to be a trend showing a potential decrease in change in GMFM88 based on the age of the patient. However, this requires further analysis to be certain.

Line 131-132, Supplementary videos.

Greater explanation is necessary with the videos.

What was being assessed pre-intervention and post-intervention for stepping on a treadmill. Shoes were worn in one video at the start and not the post-training video as child is barefoot. This could have a difference in response to the treadmill and load. Is there a way to quantify what was observed?

Response: All children were asked to take off shoes and braces before therapy. The video has been corrected. The assessment done during locomotor activity were primarily qualitative in nature. We have updated the methods to describe the assessments. The text now reads “qualitative assessment of stepping on treadmill (with minimal support and assistance, as needed) and qualitative assessment of stepping overground (with minimal support and assistance, as needed). Assessments were done on 3 criteria’s, ability to take weight bearing steps independently, to take steps with some assistance (assistance for weight bearing and/or assistance to move limbs) and being able to take steps with full assistance (assistance for weight bearing and assistance to move limbs).”

Stable assistance was provided at the trunk and pelvis that appears to be quite static and not providing dynamic movements of the pelvis/trunk consistent with stepping. The treadmill speed and 'load' (unknown) were then the two afferent inputs. Identifying the treadmill speed and the load would be valuable as well relative to delivery of this aspect of the activity-based 'neurorehabilitation' intervention?

Response: The therapy team handling the patients are experienced in working with pediatrics and provided support at the trunk and pelvis in a dynamic manner. The speed of the treadmill has been added to the methods. The weight support/load was provided as needed based on the patient’s ability and performance.

Using the term 'kid' in the video title is quite informal. Leave this to the Editors, but prefer term 'child' or 'participant' in a scientific journal.

Response: We have modified the term, it now reads child.

Line 134. Who observed or reported these findings? It is a broad, sweeping statement without explanation of how observed, when, by who, how many times, assessment. Provide detailed description

of these outcomes, e.g., parent interview, observed by clinician at what timepoints, recorded by what method, video?

Response: The parents were interviewed before every session to assess the changes since the last session. We have updated the results section to provide details regarding the changes that occurred and at what time points. The text now reads “In addition, during interviews with the study team, all parents reported that their child voluntarily began to practice the newly learned motor skills at home even in the absence of neuromodulation after two to four sessions, further enhancing neuroplasticity and increasing the child’s activity level. ”.

Line 138-149. Again, who observed these? What is 'more relaxed'? Smiled more frequently....not sure how to interpret this 'observation'.

Response: The reviewer is correct in stating that these ‘observations’ are hard to interpret. Since CP has such a wide impact of these children, and there aren’t clear metrics and instruments to measure subtle changes, the reported changes are anecdotal at best. However, we believe each of these has an impact on the quality of life of the individual and family members and contributes to the functional improvements observed in the GMFM88 scores. We have modified the text, it now reads “Further, anecdotally, parents also reported that their children demonstrated improvement in many of the symptoms secondary to cerebral palsy including improvement in visual regard and eye tracking (Fig. 3E); decreased startled response to usual stimuli such as touch (Fig. 3C), disappearance of primitive reflexes, such as asymmetrical tonic neck reflex (Fig. 3G); organized and more normal sleep patterns including being able to sleep continuously through the night; being more attentive; and increased attempts at verbal communication (Fig. 3D and 3H). A consistent observation was that all children seemed more relaxed, smiled more frequently and were able to respond to verbal requests from parents (Fig. 3H).”.

Line 140. More detailed information of the Peds QoL would be valuable to understand instead of a score change. What specifically changed for family members? Did any of the participants complete the QoL assessment? Did this complement any anecdotal findings?

Response: The PedsQL is a standard instrument to measure the change in quality of life of the family. The Pediatric Quality of Life Inventory (PedsQL) Measure- The PedsQL 4.0 GenericCore Scales were specifically designed for application to healthy children and children with chronic conditions (Varni et al. 2001). The PedsQL 3.0 CP Module was designed to measure HRQOL dimensions specific to CP. (Varni et al., 2006).

Discussion

Line 185, Suggest listing 16 sessions (if all achieved) within 8 weeks instead of 8 weeks of intervention.

Response: We have modified the text accordingly.

Appreciate the robust finding relative to the heterogeneity of the population. Understanding specific presentations would further this point, e.g., hemiplegia, spastic dyplegia.

Response: We have added the CP diagnosis to table 1.

Line 260-220. Is this statement consistent with the description of the selection of functional activities in the methods section?

Response: Sorry, we are not sure what lines the reviewer is referring to.

Can you address development across domains? Does potential change in one domain, locomotor, impact another? Developmental theories may support such findings, i.e., Adolph and Koch, 2019.

Response: Because we did not train stepping ONLY, but addressed activities that are age appropriate for all children, one domain probably did affect the other domains. Children that functioned at Level V were more willing to try higher level domains with the stimulation on. This led to more attempts at home to practice these emerging behaviors. For example: one child who initially did not have voluntary leg movements, began to kick over her iPad holder next to her bed to get her mother's attention indicating she wanted to get up out of bed. The child later was able to kick accurately at a target with her boots. This ability to flex her knee and hip led to improved and faster rolling over. Another child, who previously was static when placed on the couch, bed, or floor, started leaving his bed after his nap, and belly-crawled down a long hallway in search of his mother. Even an attempt had never occurred before. He also began to attempt to get up and down off the couch, so had to be watched more carefully. A child at Level II spontaneously began to run longer distances when taking a walk and jumping off the couch. These behaviors also had never been attempted before the study, but emerged during the study. Thus, it is likely all motor behaviors appear to be linked as they emerge, one building on the other as strength improves through trial and error or new motor behavior. Our hypothesis is based on observations from the 16 children in the study, and is validated in GMF88M test scores.

Line 239 'near normal postural and locomotor functions'. This statement may be a leap. Specific testing of these two capacities even beyond the GMFM88 may be important to make such a very strong interpretation of the current data.

Response: We have modified the statement.

In addition, 'normal' function would also mean longevity and durability, repeatability.

Response: We have deleted this sentence.

The Peds QoL outcome is not discussed. What is gained from this outcome?

Response: We have expanded the discussion to include the PedsQL

Recommend a Limitations section, e.g., durability not assessed, blinded assessors (?), skewed GMFCS levels and III not represented, what else?

What next?

Response: We have addressed the limitations in the discussion, it now reads "There are several limitations in the present study. The current study does not include GMFCS levels III and the effectiveness of the interventions for all age groups remains uncertain. While the results across this heterogenous patient cohort are consistent, unbiased assessment of outcomes with a sham arm should be a target in future trials. Further studies are urgently needed to define the frequency at which the individual needs to receive the spinal neuromodulation and ABNT in order to sustain the newly acquired motor skills. Further, the long term (over the course of several years) impact of spinal neuromodulation and ABNT on joint health, potential deformities and induced pain remains unknown.

”

Reviewer #3 (Remarks to the Author):

This is a research report of 14 children with cerebral palsy who have undergone a combination of activity-based neurorehabilitation with transcutaneous electrical spinal cord stimulation over the course of 8 weeks. The subjects were categorized using the gross motor function classification scale as ranging from level I to V. Participants were tested using the gross motor function measure-88 as the primary outcome measure both before treatment and after the 8 weeks of treatment. The abstract mentions assessing treadmill stepping and overground stepping however those results were not quantified nor reported. Anecdotal reports of improvements in other functions are mentioned as are the results of improvement in quality of life (the PedQoL survey). The authors report that all subjects responded to treatment and improved in their GMFM88 scores. It is encouraging to see that electrical stimulation is being utilized in these populations and I am hopeful for this technology. However, I have concerns outlined below:

1) It's not clear that both the activity-based training plus the transcutaneous stimulation are needed for the improvements reported here. In fact several studies show that activity-based training or goal-directed physical activity alone can lead to improvements in motor function in people with cerebral palsy (Larsen et al., 2021; Mirich et al., 2021; Zocolillo et al., 2016; Valentin-Gudiol et al., 2017; Lauruschkus et al., 2017). Were there any subjects which received only physical training or only spinal stimulation? It's difficult to draw conclusions that the transcutaneous stimulation is necessary or even helpful in this context.

Response: Most of the references listed here focused on either adults with CP, or included other means of rehab (such as VR) in the activity based rehab. In Valentin-Gudiol et al., 2017, they reported a meta-analysis of only 5 studies since 2011 for all forms of pediatric developmental delays and stated that "This update of the review from 2011 provides additional evidence of the efficacy of treadmill intervention for certain groups of children up to six years of age, but power to find significant results still remains limited. The current findings indicate that treadmill intervention may accelerate the development of independent walking in children with Down syndrome and may accelerate motor skill attainment in children with cerebral palsy and general developmental delay."

Thus, the effect of training alone seems to be inconclusive at best.

In our experience working with adults with a fully developed nervous system that is dysfunctional due to a SCI, training alone had little impact on improving voluntary sensorimotor function. It wasn't until neuromodulation (epidural or transcutaneous) was introduced that patients were able to regain voluntary control.

The objective of the current first in human study was not to identify the differences in responses to ABNT and spinal neuromodulation, rather to demonstrate the improvements possible with a combination of the two in children with CP.

2) Who performed the GMFM88 scoring? Was it blinded or done with an outside clinician for an unbiased assessment?

Response: The GMFM88 assessment was performed by a member of the research team, an experienced pediatric therapist. Since this was a first in human study, it was unblinded.

REVIEWER COMMENTS

Reviewer #1 (Remarks to the Author):

While the authors have provided adequate replies to some of my comments, there are some major issues outstanding, as follows:

1) The authors must acknowledge that their study cannot separate the effects of the ABNT from those of ABNT+spinal stim.

In their reply to my previous comment on this point, the authors stated that “The current standard of care (SoC), i.e., PT/OT has shown limited success in improving functional movement. The improvements in GMFM88 scores ranged from 1.3 to 6.2 points after 3 weeks to 8 months of SoC PT (Bakanjee et al., 2018).”

However, their ABNT was not standard of care, it was a “novel combination of activity-based neurorehabilitation therapy (ABNT)” (line 64), which was designed “to optimize proprioceptive information being processed by the nervous system” (line 140-141). There is strong evidence from animal studies that proprioceptive feedback is essential for the recovery of function (after spinal cord injury) [1]. Therefore, there is (at least) a reasonable possibility that the novel ABNT alone elicited the effects noted. I understand that the objective of the study was not to study the differences between ABNT and spinal neuromodulation, but the authors must still acknowledge the point that the novel ABNT therapy alone has never been studied in this population and they cannot rule out the possibility that it would have been effective given alone.

2) In response to my previous comment about stimulation intensity, the authors stated that they set the intensity at “20% below lowest lower extremity muscle motor threshold” (line 102). The authors need to provide information in the methods about how they measured motor threshold, and the data should be provided in the results. This would be novel data, which would strengthen the paper considerably.

3) The authors need to check their data for normality and perform their statistics accordingly.

Specific points:

Abstract

Line 12: change “we hypothesized that” to “we explored whether”

Lines 19-21: the modified sentence is satisfactory, except that GMFM88 needs to be defined.

“We disagree, however, with the Reviewer’s viewpoint which assumes that reorganization of networks can be demonstrated by only neurophysiology and imaging, when in fact neither is a more direct demonstration than the changes in EMG patterns which is a direct functional demonstration in vivo conditions, not a “biomarker” of reorganization. We have demonstrated these results in our previous study which is the basis for the current study.”

My point was that you cannot state: “We demonstrate that transcutaneous spinal neuromodulation during ABNT can transform the neural networks in children diagnosed with CP” in the abstract of a study that did not use any EMG/neurophysiology, because the point is not supported by the results presented in the current study. “We have previously demonstrated.....” with the relevant reference would be more appropriate.

Line 54: I cannot find a definition of GMFM88 in the Introduction, and no references to support its validation in children with CP have been provided, as requested in my first review. Be consistent in your use of GMFM88 vs GMFM-88.

Line 57: this abbreviation SCI appears before its definition (provided in line 63)

Lines 82-83: Please could the authors justify the lack of upper age limit. The study Title refers to “children with CP” yet their inclusion criteria means that adults with CP are also eligible to participate.

Lines 73-74: “We hypothesized that, spinal neuromodulation and appropriate proprioception during ABNT, can facilitate learning of sensorimotor function in children with CP.”

I do not consider this to be an appropriate hypothesis. A hypothesis should be measurable and it is unclear how “learning of sensorimotor function” is quantified.

Lines 74-77: I don't follow this sentence. There is a typo “with the Gross Thus”, which might help clarify the sentence when corrected.

In my opinion, the hypothesis and objective should be switched, as follows: “The objective of our study was to determine the effect of spinal neuromodulation and appropriate proprioception during ABNT on sensorimotor function in children with CP. We hypothesized that ABNT in the presence of spinal neuromodulation in children with CP would increase GMFM88 scores.”

Line 88: the use of the word “biological” in this sentence is ambiguous.

Line 99: As requested in my previous review, please state whether the stimulation is synchronous.

Line 102: “20% below lowest lower extremity muscle motor threshold” Please provide details about how (and how often) motor threshold was measured. Was EMG used for this? Provide details in methods. Also provide the motor threshold data for each lower limb muscle measured in your results section.

In our experiments (in adults) using pulses modulated with a 10kHz frequency, the lowest motor threshold (based on measurement of PRRs) occurred at ~160-200mA. 20% below this would be 128-140mA, which is an order of magnitude higher than the 10-18mA used here. It is essential that the authors provide their motor threshold data to substantiate the selected intensity.

In addition, providing details about the determination of motor threshold would also validate their electrode positioning. While the target (e.g. vertebral level T11/T12) is based on their previous experience, human anatomy is variable. Verifying that the electrodes are positioned appropriately over the targeted motor pools by eliciting motor responses (PRRs) is standard and the author's validation of electrode placement needs to be described fully in the methods.

Line 157: the link to the reference is broken.

Line 162: Criteria is a plural noun (its singular is criterion)

Lines 168-169: The authors must test their data for normality and perform statistics accordingly, particularly considering the relatively low number of subjects in the trial.

Lines 193-195: I'd suggest the authors remove this statement as it is not supported by the data presented in Fig. 2C. Generally, an R-squared value <0.3 is considered a low or weak association; the value reported in Fig 2C is 0.08.

References

1. Takeoka, A., I. Vollenweider, G. Courtine, and S. Arber, Muscle spindle feedback directs locomotor recovery and circuit reorganization after spinal cord injury. *Cell*, 2014. 159(7): p. 1626-39.

Reviewer #2 (Remarks to the Author):

Appreciate the revisions and addition of the limitations section and inclusion of a sham group. Suggest adding the lack of a blinded assessor to the limitations section and thus blinded assessors needed for future work.

Suggest adding the individual changes in the GMFM-88 scores for the individual 16 across the levels of GMFCS in supplementary material. The overall score is valued, however knowledge of 'how' children advanced across levels of the GMFM-88 is also valuable noting that children in Level IV and V will have different motor deficits than children in Levels I and II.

Line 67 - please provide the reference for WHO guidelines that describe ABNT....

Line 155 - note reference error

Line 160. Suggest re-wording this statement; such as "Assessments addressed three specific criteria..."

Line 196. The use of the term 'non-ambulatory' remains unclear. Is this description specific to 'initiating and taking steps on the treadmill' or does it refer to a child's ability to independently walk over ground. Suggest reporting the ability to walk also based on the 16 children noting their initial GMFCS categorization. A child who is initially a GMFCS V and non-ambulatory and becomes 'ambulatory' is a significant outcome if 'ambulatory' refers to walking overground with reciprocal steps even with a walker. Please clarify further.

Reviewer #3 (Remarks to the Author):

With utmost respect to the authors and for the work undertaken in this study, I'm not yet convinced that this is suitable for publication at this time. My concerns are as follows:

- 1) It is still problematic for me that the two therapies (ABNT and transcutaneous spinal stim) are not separated. The authors state in their rebuttal letter that the previous papers showing improvement in motor function with ABNT alone show lower (or more variable) scores than that reported here. However this could be a result of a particularly successful ABNT therapy here, despite the best efforts to state that the stimulation is responsible. In addition, though the authors freely disclosed the lack of controls as a limitation, the combination of all these factors makes it very difficult to assess unbiased conclusions.
- 2) The authors also state in the rebuttal letter that the children are learning new tasks and performing them at home. It's difficult since those are qualitatively reported by the caregivers/parents and not quantitatively assessed.
- 3) Behavioral assessments for neurologic function are notoriously difficult in that they can be unwittingly biased. This is why multiple, blinded assessments are done. I believe having one unblinded assessor for both pre- and post-therapies along with the self-reporting (from caregivers/parents) of qualitative improvements at home makes this harder to validate.

Reviewer #1 (Remarks to the Author):

While the authors have provided adequate replies to some of my comments, there are some major issues outstanding, as follows:

1) The authors must acknowledge that their study cannot separate the effects of the ABNT from those of ABNT+spinal stim.

In their reply to my previous comment on this point, the authors stated that “The current standard of care (SoC), i.e., PT/OT has shown limited success in improving functional movement. The improvements in GMFM88 scores ranged from 1.3 to 6.2 points after 3 weeks to 8 months of SoC PT (Bakanjee et al., 2018).”

However, their ABNT was not standard of care, it was a “novel combination of activity-based neurorehabilitation therapy (ABNT)” (line 64), which was designed “to optimize proprioceptive information being processed by the nervous system” (line 140-141). There is strong evidence from animal studies that proprioceptive feedback is essential for the recovery of function (after spinal cord injury) [1]. Therefore, there is (at least) a reasonable possibility that the novel ABNT alone elicited the effects noted. I understand that the objective of the study was not to study the differences between ABNT and spinal neuromodulation, but the authors must still acknowledge the point that the novel ABNT therapy alone has never been studied in this population and they cannot rule out the possibility that it would have been effective given alone.

Response: We have updated the manuscript to acknowledge the objective was to study the combined effect and have acknowledged this being a potential limitation of the study.

2) In response to my previous comment about stimulation intensity, the authors stated that they set the intensity at “20% below lowest lower extremity muscle motor threshold” (line 102). The authors need to provide information in the methods about how they measured motor threshold, and the data should be provided in the results. This would be novel data, which would strengthen the paper considerably.

Response: We have updated the manuscript to further explain the protocol used to determine thresholds and the intensities used during training. The reference to the lower extremity muscle was from our previous study (Gad et al., 2021). The method used in the current study has been clarified in the methods section. We have also included the ranges of thresholds for each site.

3) The authors need to check their data for normality and perform their statistics accordingly.

Response: All data were tested for normality using the Kolmogorov-Smirnov test. Based on the result of a normal distribution, we used the paired t-test to test to assess statistical significance. We have updated the methods accordingly.

Specific points:

Abstract

Line 12: change “we hypothesized that” to “we explored whether”

Response: Updated as suggested

Lines 19-21: the modified sentence is satisfactory, except that GMFM88 needs to be defined.

Response: We have defined GMFM-88 in the abstract

“We disagree, however, with the Reviewer’s viewpoint which assumes that reorganization of networks can be demonstrated by only neurophysiology and imaging, when in fact neither is a more direct demonstration than the changes in EMG patterns which is a direct functional demonstration in in vivo conditions, not a “biomarker” of reorganization. We have demonstrated these results in our previous study which is the basis for the current study.”

My point was that you cannot state: “We demonstrate that transcutaneous spinal neuromodulation during ABNT can transform the neural networks in children diagnosed with CP” in the abstract of a study that did not use any EMG/neurophysiology, because the point is not supported by the results presented in the current study. “We have previously demonstrated.....” with the relevant reference would be more appropriate.

Response: I believe we have taken out that sentence from the previous submission. Could the reviewer please identify the line number they are referring to.

Line 54: I cannot find a definition of GMFM88 in the Introduction, and no references to support its validation in children with CP have been provided, as requested in my first review. Be consistent in your use of GMFM88 vs GMFM-88.

Response: The definition of GMFM-88 has been added. GMFM-88 is now referred to as GMFM-88 throughout the document.

Line 57: this abbreviation SCI appears before its definition (provided in line 63)

Response: The full form of SCI has been updated on the first occasion.

Lines 82-83: Please could the authors justify the lack of upper age limit. The study Title refers to “children with CP” yet their inclusion criteria means that adults with CP are also eligible to participate.

Response: The original study was designed as a proof of principle and also included adults with CP.

However, the individuals that were recruited in this study were children under the age of 18. Thus, there is no upper limit mentioned (consistent with the information on the clinicaltrials.gov listing).

Lines 73-74: “We hypothesized that, spinal neuromodulation and appropriate proprioception during ABNT, can facilitate learning of sensorimotor function in children with CP.”

I do not consider this to be an appropriate hypothesis. A hypothesis should be measurable and it is unclear how “learning of sensorimotor function” is quantified.

Response: In line with the comment below, we have updated the text as suggested.

Lines 74-77: I don’t follow this sentence. There is a typo “with the Gross Thus”, which might help clarify the sentence when corrected.

Response: We have deleted the sentence

In my opinion, the hypothesis and objective should be switched, as follows: “The objective of our study was to determine the effect of spinal neuromodulation and appropriate proprioception during ABNT on sensorimotor function in children with CP. We hypothesized that ABNT in the presence of spinal neuromodulation in children with CP would increase GMFM88 scores.”

Response: We have updated the text as suggested.

Line 88: the use of the word “biological” in this sentence is ambiguous.

Response: We have changed the term to “neurophysiological”.

Line 99: As requested in my previous review, please state whether the stimulation is synchronous.

Response: If we understand this comment, the reviewer is asking if the stimulation delivered at the two sites occurs at the same time, i.e., simultaneously. If our interpretation is correct, the stimulation is always delivered at the two sites simultaneously. This is consistent with our previous studies.

Line 102: “20% below lowest lower extremity muscle motor threshold” Please provide details about how (and how often) motor threshold was measured. Was EMG used for this? Provide details in methods. Also provide the motor threshold data for each lower limb muscle measured in your results section.

Response: As stated in the methods, motor thresholds were assessed visually and the stimulation intensity during training were set according to the muscle response observed at the lowest intensity. We have updated the methods section, providing more detail on how the stimulation intensities were determined. These responses are consistent with our previous study that used EMG as a biomarker to identify stimulation intensities (Gad et al., 2021).

In our experiments (in adults) using pulses modulated with a 10kHz frequency, the lowest motor threshold (based on measurement of PRRs) occurred at ~160-200+mA. 20% below this would be 128-140mA, which is an order of magnitude higher than the 10-18mA used here. It is essential that the authors provide their motor threshold data to substantiate the selected intensity.

Response: There are a few fundamental differences between the methods stated by the reviewer (assuming the reference to be Al'joboori, Y. et al., 2021) and our approach:

- 1) The thresholds in adults and children are expected to be different. Also, stimulation at these levels would certainly trigger negative responses more frequently for children and their parents. But from a logical experimental perspective, the amps in the range of ~160-200+mA is completely counter to our concept of enabling as opposed to induces motor behavioral as we have noted in many previous publications.*
- 2) The method used to determine threshold while monitoring the PRRs use single (or double) pulse(s) rather than a continuous train at 30Hz*
- 3) The pulses used to measure PRRs were monophasic compared to biphasic used in this study*

The TA muscle was used as a reference to determine motor threshold rather than visual extension of cervical or thoracic regions. We would question the scientific evidence for depending on the response of the TA to be the best marker of a relative physiological response?

(see Spinal Cord. 2005 Jan;43(1):14-21. doi: 10.1038/sj.sc.3101656. Reflex reciprocal facilitation of antagonist muscles in spinal cord injury. R Xia¹, W Z Rymer)

Thus, with these key fundamental differences, it is not scientifically valid to compare our methods and results with the reviewer's approach. Further, the objective of this study was to determine the chronic effects of functional spinal neuromodulation and ABNT. We have updated the methods section to list the range of motor threshold intensities and the intensities used during training for each site.

In addition, providing details about the determination of motor threshold would also validate their electrode positioning. While the target (e.g. vertebral level T11/T12) is based on their previous experience, human anatomy is variable. Verifying that the electrodes are positioned appropriately over the targeted motor pools by eliciting motor responses (PRRs) is standard and the author's validation of electrode placement needs to be described fully in the methods.

Response: While we agree with the reviewer that variability may exist across different subjects, the objective of this study was not to identify the most optimal stimulation site for every individual but rather to identify the effect of ABNT and spinal neuromodulation as to how this response relates to changes in GMFM-88 scores. We can't assume that our stim levels are optimal in generating a chronic effect. The most effective procedure was to be consistent in selecting parameters as noted in the revised methods for each subject.

Line 157: the link to the reference is broken.

Response: We have verified the reference via endnote.

Line 162: Criteria is a plural noun (its singular is criterion)

Response: We have corrected the typo

Lines 168-169: The authors must test their data for normality and perform statistics accordingly, particularly considering the relatively low number of subjects in the trial.

Response: All data were tested for normality using the Kolmogorov-Smirnov test. Based on the result of a normal distribution, we used the paired t-test to test to assess statistical significance. We have updated the methods accordingly.

Lines 193-195: I'd suggest the authors remove this statement as it is not supported by the data presented in Fig. 2C. Generally, an R-squared value <0.3 is considered a low or weak association; the value reported in Fig 2C is 0.08.

Response: The R-squared values were included based on the recommendations from the previous review. We have dropped the R-squared value.

References

1. Takeoka, A., I. Vollenweider, G. Courtine, and S. Arber, Muscle spindle feedback directs locomotor recovery and circuit reorganization after spinal cord injury. *Cell*, 2014. 159(7): p. 1626-39.

Reviewer #2 (Remarks to the Author):

Appreciate the revisions and addition of the limitations section and inclusion of a sham group. Suggest adding the lack of a blinded assessor to the limitations section and thus blinded assessors needed for future work.

Response: We have included the need for blinded assessment in the limitation section.

Suggest adding the individual changes in the GMFM-88 scores for the individual 16 across the levels of GMFCS in supplementary material. The overall score is valued, however knowledge of 'how' children advanced across levels of the GMFM-88 is also valuable noting that children in Level IV and V will have different motor deficits than children in Levels I and II.

Response: We have added supplementary Table 1 that list the individual scores for the 16 children.

Line 67 - please provide the reference for WHO guidelines that describe ABNT....

Response: The reference has been added.

Line 155 - note reference error

Response: The reference has been corrected via endnote.

Line 160. Suggest re-wording this statement; such as "Assessments addressed three specific criteria..."

Response: The sentence has been updated as recommended.

Line 196. The use of the term 'non-ambulatory' remains unclear. Is this description specific to 'initiating and taking steps on the treadmill' or does it refer to a child's ability to independently walk over ground. Suggest reporting the ability to walk also based on the 16 children noting their initial GMFCS categorization. A child who is initially a GMFCS V and non-ambulatory and becomes 'ambulatory' is a significant outcome if 'ambulatory' refers to walking overground with reciprocal steps even with a walker. Please clarify further.

Response: We have updated the results to describe the use of the term non-ambulatory and provided details as to the responses recorded to the neuromodulation-training relative to the GMFCS levels.

Reviewer #3 (Remarks to the Author):

With utmost respect to the authors and for the work undertaken in this study, I'm not yet convinced that this is suitable for publication at this time. My concerns are as follows:

1) It is still problematic for me that the two therapies (ABNT and transcutaneous spinal stim) are not separated. The authors state in their rebuttal letter that the previous papers showing improvement in motor function with ABNT alone show lower (or more variable) scores than that reported here. However this could be a result of a particularly successful ABNT therapy here, despite the best efforts to state that the stimulation is responsible. In addition, though the authors freely disclosed the lack of controls as a limitation, the combination of all these factors makes it very difficult to assess unbiased conclusions.

Response: We have updated the limitation and stated that ABNT alone could have an impact on its own and needs to be explored in the future. This perception is certainly a common one, and we understand that due to this commonality of this opinion that it is reasonable or even essential and therefore is not publishable. But we are confident that this perception is contrary to our basic biology, although satisfying from the viewpoint of attempting to identify specific functions with a specific and highly isolated experimental conditions. This steadfast viewpoint completely ignores the basic designs of the nervous system, which is to accommodate an effective integration of multiple physiological systems. We are not aware, at least at the systems level, which provides any evidence that an attempt to separate the relative effects of the two phenomena as complicated as electrical spinal neuromodulation and the recovery of function of highly complex voluntary sensorimotor events is even conceptually combatable with our basic biology. A good example of this is demonstrated in publications in which we have tested two different interventions after a spinal cord injury, both of which had been demonstrated to be effective in improving function, but when the two were combined, the level of recovery was not significantly different from the group of rats that received neither intervention (Maier et al., Brain 2009). Essentially, to attempt to dissect the relative effects, even of two different very simple interventions, much less ones that are as complex as an intervention to manipulate the efficacy of No-go and the highly systemic control mechanisms involved in performing a highly complex sensory-motor function, which involves a step-by-step and millisecond to millisecond control of motor pools that control a majority of the motor pools involved in full weight bearing stepping under in vivo conditions. We are unaware of any convincing evidence that is contrary to our stated position that to attempt to separate the effects of the interventions being combined is even logical from a basic biology perspective. If there is such evidence, we would appreciate being informed as such. From a

systemic biologist's perspective, we are of the opinion that the more reduced state of a biological experimental model being used, the more irrelevant the experimental results as it relates to normal in vivo biology. This is not to say at all, of a limited importance of experimentally reduced models to the most elemental components, but there has been an increasing frequency of the interpretation of the more reduced experiments that ignores its limitations with respect to basic biological principles as related to integration functions.

2) The authors also state in the rebuttal letter that the children are learning new tasks and performing them at home. It's difficult since those are qualitatively reported by the caregivers/parents and not quantitatively assessed.

Response: We agree with the reviewer, these changes are anecdotal in nature. However, since there is no single instrument that can measure these changes, we have only presented our observations. This is an area that has been generally ignored and represents an area of major limitations in assessing the effects of an intervention that last for, perhaps three or four hours in a week while the rest of the patients' time outside of the clinic, is completely unaccounted for. Given the massive amount of data demonstrating the effects of use dependent plasticity as a mechanism for recovery of function, it is almost certain that future studies that will provide quantitative data on the behavior of the subject outside the clinic will become more and more important. In the present case with CP, it has become evident to us repeatedly that this is an area that definitely needs to be corrected. This has become extremely evident with children with CP, because of the immediate effect of neuromodulation in broadening the number of activities that the children seem highly motivated to perform even without direction and encouragement. Basically, what we observe with the neuromodulation, they gain a sense of being able to perform a broader range of activities and they are highly motivated to explore these possibilities. On the other hand, in the same intervention session, we rarely if ever see such acute responses after training in the absence of neuromodulation. Consequently, since this is a first in human study and the improvements observed here are far greater than what has been reported earlier, we feel it is imperative that we mention them, not as proof, but of something that is too suggestive, to be ignored at the onset. We have acknowledged that these observations are anecdotal and require more in-depth assessments of potential mechanisms and clear clinical results using more sensitive instruments.

3) Behavioral assessments for neurologic function are notoriously difficult in that they can be unwittingly biased. This is why multiple, blinded assessments are done. I believe having one unblinded assessor for both pre- and post-therapies along with the self-reporting (from caregivers/parents) of qualitative improvements at home makes this harder to validate.

Response: Since this is our first in human study, we have only blinded the parents. We have acknowledged the need for blinded assessments including a sham arm in future larger clinical trials to test the chronic effects of these two interventions for a large but specified segment of CP and related patients.

This issue of blinding a study with children that are severely impaired is definitely a limitation in the present observations. However, recognition of this limitation is indeed important, but we are confident that the magnitude of the changes seen with neuromodulation and the activity-based training has resulted in improvements in function that are significantly greater than what has been reported before, both in the magnitude of the changes as well as how rapidly they can occur when both interventions are applied. But another factor that motivates us in this area is that well trained and experienced pediatric therapists and the patients' parents consistently point out that they have rarely if ever seen such improvement occur in individuals with severe dysfunction due to CP. But the other underlying motivation is that there are several surgical procedures that are becoming routine that can have some relatively temporary benefits, but the

long-term potential for further improvement as they grow older, is severely limited. We feel that it is urgent to have sufficient data to at least provide caution as to whether parents with children as young as two years of age are being encouraged to receive these invasive irreversible surgical interventions. We consider it to be a highly ethical position to at least inform those with these conditions, be aware of potential alternatives although the efficacy of these alternatives that show unusual promise need further and more critical evaluations that will stand the test of time. We judge this position to be ethically, the most appropriate strategy forward.

REVIEWER COMMENTS

Reviewers' comments:

Reviewer #1 (Remarks to the Author):

I would like to thank the authors for their replies to my previous comments and modifications to their manuscript. Please see some further comments below. The main point requiring further clarification is regarding definition of motor threshold.

Line 43: There is a typo

Lines 112-113: "the motor threshold was defined as the amplitude at which the child first attempted to extend the cervical or thoracic regions while in a seated position"

This is not how motor threshold is typically defined, and does not agree with the previous publication the authors refer to (Gad et al. 2021). Motor threshold is usually the lowest level of stimulation that elicits a (monosynaptic or direct) motor response. Indeed, as stated in their previous publication, "the stimulation intensity was maintained at ~ 20–25% below the threshold that induced a motor response" (Gad et al., 2021, p1955).

The reported stimulation parameters in Gad et al 2021 were in the region 15-20mA only for children 2 years of age. The remaining received stimulation in the range 25-40mA, which is more than double the values reported here (10-16mA). It is important for the authors to clarify their chosen approach for defining motor threshold. In particular, how was an "attempt to extend the cervical or thoracic regions" defined? What instructions were given to the participants? If the point was to find an "enabling" stimulation intensity, then why provide stimulation at 20% lower? I would also recommend placing the stimulation parameters used in Table 1, as was done in their previous publication.

Line 166: link to reference is broken

Lines 203-206 & Fig 2C: The authors may have misunderstood my point. The R squared values were helpful and should remain in Figure 2C. Lines 203-206 should be removed. Here, the authors suggest that an association between age and GMFM-88 scores exists. This statement is not supported by their data because the association is weak (as indicated by the low R squared value).

Line 286: there is a typo

Reviewer #2 (Remarks to the Author):

Line 166 reference remains in 'error'.

Appreciate the reviewers' sufficient response in addressing prior comments including GMFM-88 and additional table of individual scores, defining ambulatory, and the limitation section expansion.

Reviewer #3 (Remarks to the Author):

With all due respect, I don't believe this study meets the standards to be published in this journal. Given the unblinded nature of the outcome measures, the anecdotal reports of behavioral improvements, and the fact that both activity-based rehabilitation (published previously and shown to be successful in subjects with cerebral palsy) and electrical stimulation are given at the same time,

there's no clear answer that the authors' experimental set-up did indeed improve motor outcomes in the way they are claiming.

The author's response to my questions included the statement, 'We have acknowledged that these observations are anecdotal and require more in-depth assessments of potential mechanisms and clear clinical results using more sensitive instruments.' while also stating, '..since there is no single instrument that can measure these changes, we have only presented our observations.' There are, in fact instruments to measure motor and sensory outcomes in clinical settings (e.g. electromyography, motor evoked- and somatosensory evoked potential testing, etc.). It's more of a fundamental issue with experimental design and rigor that keeps me from accepting the study as is. That being said, I genuinely appreciate the time and effort that the authors have taken in this endeavor and definitely encourage them to consider these limitations in the design of future studies because this is very important work.

Reviewer #1 (Remarks to the Author):

I would like to thank the authors for their replies to my previous comments and modifications to their manuscript. Please see some further comments below. The main point requiring further clarification is regarding definition of motor threshold.

Line 43: There is a typo

Response: The typo has been corrected.

Lines 112-113: “the motor threshold was defined as the amplitude at which the child first attempted to extend the cervical or thoracic regions while in a seated position”

This is not how motor threshold is typically defined, and does not agree with the previous publication the authors refer to (Gad et al. 2021). Motor threshold is usually the lowest level of stimulation that elicits a (monosynaptic or direct) motor response. Indeed, as stated in their previous publication, “the stimulation intensity was maintained at ~ 20–25% below the threshold that induced a motor response” (Gad et al., 2021, p1955).

The reported stimulation parameters in Gad et al 2021 were in the region 15-20mA only for children 2 years of age. The remaining received stimulation in the range 25-40mA, which is more than double the values reported here (10-16mA). It is important for the authors to clarify their chosen approach for defining motor threshold. In particular, how was an “attempt to extend the cervical or thoracic regions” defined? What instructions were given to the participants? If the point was to find an “enabling” stimulation intensity, then why provide stimulation at 20% lower? I would also recommend placing the stimulation parameters used in Table 1, as was done in their previous publication.

Response: *We acknowledge that the identification of stimulation intensities was done without the use of EMG as done in our previous publication. Three key differences exist in the current study and our previous study, 1) Gad et al., 2021 focused only on the ability to step on a treadmill whereas the current study focuses on changes in global sensorimotor function as measured by the GMFM-88 instrument, 2) Since Gad et al., 2021 was focused on stepping only, the stimulation electrodes were placed at T11-12 and L1-2 vertebral levels whereas in the current study the stimulation electrodes were placed at C5-6 and T11-12. Further, the threshold for eliciting a motor response is lower with cervical stimulation compared to thoracic, thus the intensities used were much lower compared to Gad et al., 2021 and 3) This study was designed to observe the functional improvements on the GMFM-88 total scores and not the electrophysiological changes that occur with vs without stimulation. Thus, all assessments were done in the absence of stimulation.*

The children were not given any instructions when identifying the stimulation parameters. The therapist delivering the stimulation observed for cues of motor responses which determined threshold. These were also verified by video recording. We followed our usual enabling phenomenon and thus further lowered the intensity by 20% during training. We have updated table 1 to list the thresholds for each patient. We have modified the methods to provide further details.

Line 166: link to reference is broken

Response: *We apologize for the error. The link has now been fixed.*

Lines 203-206 & Fig 2C: The authors may have misunderstood my point. The R squared values were helpful and should remain in Figure 2C. Lines 203-206 should be removed. Here, the authors suggest

that an association between age and GMFM-88 scores exists. This statement is not supported by their data because the association is weak (as indicated by the low R squared value).

Response: *We have removed the lines as suggested and re-added the R squared values.*

Line 286: there is a typo

Response: *We apologize, we cannot find the typo on line 286. The text currently reads "It appears that practicing motor tasks with incorrect alignment and posture could lead to enhancement aberrant connections within and among the brain and spinal neural networks."*

Reviewer #2 (Remarks to the Author):

Line 166 reference remains in 'error'.

Response: *We apologize for the 'error' in the reference. It has now been fixed.*

Appreciate the reviewers' sufficient response in addressing prior comments including GMFM-88 and additional table of individual scores, defining ambulatory, and the limitation section expansion.

Response: *We appreciate the feedback.*

Reviewer #3 (Remarks to the Author):

With all due respect, I don't believe this study meets the standards to be published in this journal. Given the unblinded nature of the outcome measures, the anecdotal reports of behavioral improvements, and the fact that both activity-based rehabilitation (published previously and shown to be successful in subjects with cerebral palsy) and electrical stimulation are given at the same time, there's no clear answer that the authors' experimental set-up did indeed improve motor outcomes in the way they are claiming.

Response: *We respectfully disagree with the reviewers comment about the data not being worthy of being published in Nature Communication. Keller et al., 2021 was recently published in Nature Comm as a non-blinded, non-randomized pilot study with 8 children after SCI studying the acute effect of transcutaneous spinal stimulation on upright posture over the course of only 3 days. This study did not report any electrophysiology data and used the Center of pressure recordings from the force plate and kinematics as the primary outcomes. Thus, we respectfully disagree this comment that our data are not worthy of being published in Nat Comm considering a recent paper (Kelle et al., 2021) was published using the same experimental design with fewer subjects, fewer outcomes, no training data and no clinically relevant outcomes.*

Comment: Activity based neuro rehab in CP

Response: *While activity based neurorehab has shown some improvement in function, the changes observed in the GMFM-88 scores are significantly greater in our study within a shorter period of time. For example, Ko et al., 2014, reported that GMFM-88 total after 6 months of physical therapy, 3x/week, ranged from 7.3 – 11.0 points across the GMFCS levels. While the improvements reported by Ko et al., (2014) are clinically significant (>5 points), the data from the current study demonstrates higher levels of efficacy with a change in GMFM-88 score of 12.54 despite being recorded at shorter timeframes (2 months). Thus, we do believe that the data are consistently better than previously reported literature. We*

do believe that the significant improvement in GMFM-88 score in every child tested (16/16) is due to the presence of spinal neuromodulation being delivered during ABNT.

Thus, while the data reported here are from an unblinded and single arm study, they demonstrate how a novel intervention can be developed into the new standard of care therapy for kids of CP and are in line with the data presented in previous Nat Comm Publications (Keller et al., 2021) and thus worthy of being published in this journal.

The author's response to my questions included the statement, 'We have acknowledged that these observations are anecdotal and require more in-depth assessments of potential mechanisms and clear clinical results using more sensitive instruments.' while also stating, '..since there is no single instrument that can measure these changes, we have only presented our observations.' There are, in fact instruments to measure motor and sensory outcomes in clinical settings (e.g. electromyography, motor evoked- and somatosensory evoked potential testing, etc.). It's more of a fundamental issue with experimental design and rigor that keeps me from accepting the study as is. That being said, I genuinely appreciate the time and effort that the authors have taken in this endeavor and definitely encourage them to consider these limitations in the design of future studies because this is very important work.

Response: *We respectfully disagree with the reviewers' comment. As stated in the introduction, the objective of the study was to demonstrate changes in GMFM-88 total scores and changes in locomotor capabilities. GMFM-88 is the current gold standard for sensorimotor function in children with CP and is used worldwide, thus is translatable and allows us to maintain the required rigor. The other additional changes reported varied from individual to individual and occurred at different physical locations (some at home, some in the clinic and some at school). Further, considering the wide range of observations from decreased startled response to improvement in sleep patterns to changes in asymmetric tonic neck reflex, no single instrument (electrophysiological or clinical) can quantify these changes. Identifying individual instruments to study each of these was not the primary objective of the study and will need to be individually evaluated in future studies.*

Keller A, Singh G, Sommerfeld JH, King M, Parikh P, Ugiliweneza B, D'Amico J, Gerasimenko Y, Behrman AL. Noninvasive spinal stimulation safely enables upright posture in children with spinal cord injury. *Nature communications*. 2021 Oct 6;12(1):1-1.

Ko, J., 2014. Sensitivity to functional improvements of GMFM-88, GMFM-66, and PEDI mobility scores in young children with cerebral palsy. *Perceptual and motor skills*, 119(1), pp.305-319.